# FEDERATED UNLEARNING WITH GRADIENT SHIELD-ING

## ABSTRACT

Federated unlearning enables the removal of a specific client's data contribution from a trained federated model, thereby avoiding the substantial computational cost of complete retraining. However, existing methods suffer from high memory overhead, training instability, and performance degradation on remaining clients, particularly in non-IID settings. These challenges arise from fundamental issues including gradient explosion and the conflict between forgetting and retaining gradients. To address these limitations, we propose Federated Unlearning with GrAdient Shielding (FUGAS), which integrates a novel forgetting loss with a flexible gradient projection to achieve efficient unlearning while preserving model utility, all without storing extensive historical information. Specifically, we formulate unlearning as a preference optimization problem. The model's original predictions on the data to be forgotten serve as a negative reference, and our objective function encourages the model's current outputs to diverge from this reference, effectively erasing the targeted knowledge. Concurrently, during the server aggregation phase, gradients from unlearning clients are projected onto a dynamically estimated compatibility subspace derived from the gradients of retained clients, which ensures directional coherence and mitigates destructive interference between competing updates. Furthermore, we provide theoretical guarantees that our novel forgetting loss prevent gradient explosion, and that the projection ensures a non-increase in risk on the retained tasks. Extensive experiments demonstrate that FUGAS not only achieves thorough unlearning but also consistently maintains or even improves the model's accuracy on retained data.

## 1 INTRODUCTION

Federated Learning (FL) (McMahan et al., 2017; Yang et al., 2019; Kairouz et al., 2021; Li et al., 2020a;b; T. Dinh et al., 2020; Smith et al., 2017) is a distributed machine learning framework that enables multiple clients to collaboratively train a global model without sharing their local datasets. However, the practical FL systems necessitates managing the dynamic lifecycle of data and client participation. A critical requirement in this context is Federated Unlearning (FU) (Liu et al., 2021; Gong et al., 2021a; Wu et al., 2022c; Liu et al., 2025a), which aims to remove the contributions of specific clients or data samples from a trained global model.

Current FU methods predominantly fall into two categories including update-storage and gradient-modification (Liu et al., 2025b; Dhasade et al., 2023; Halimi et al., 2022). Although conceptually straightforward, these methods face significant practical challenges. Update-storage approaches incur high memory consumption, as the storage overhead scales linearly with the number of training rounds. While memory-free gradient-modification solutions avoid this cost, they introduce the risk of training instability. Maximizing the loss on the forgetting set can lead to gradient explosion since losses like cross-entropy are unbounded above, causing drastic and unstable parameter updates (Romandini et al., 2025; Alam et al., 2024; Li et al., 2023). Furthermore, a fundamental challenge in FU arises from the inherent conflict between gradients from the forgetting and retained clients. Updates intended to erase specific data often directly oppose the updates required to maintain performance on the remaining data (Alam et al., 2024). This gradient conflict is severely exacerbated in FL scenarios with non-independent and identically distributed (non-IID) data. Consequently, existing methods face a difficult trade-off in which unlearning too cautiously fails to completely remove the targeted

influence, while unlearning too aggressively causes irreparable damage to the model's utility for the remaining clients.

To address these limitations, we propose Federated Unlearning with GrAdient Shielding (FUGAS), a novel method that achieves effective forgetting while simultaneously ensuring the performance and stability of the resulting model. Our approach introduces an integrated solution that combines a bounded forgetting objective with a flexible gradient projection mechanism. This design directly addresses the challenges of gradient explosion and gradient conflict without requiring extensive historical data storage, thus enabling effective, stable, and memory-efficient unlearning.

Specifically, we reformulate unlearning as a preference optimization problem, which treats the model's original predictions on the data to be forgotten as a negative reference (Rafailov et al., 2023). Our novel objective function encourages the model's current outputs to diverge from this reference, effectively erasing the targeted knowledge without resorting to unbounded loss maximization. To resolve the issue of gradient conflict, we employ a flexible gradient projection during server aggregation. The gradients from unlearning clients are projected onto a compatibility subspace, which is estimated from the gradients of the retained clients. Unlike prior methods that project onto a strictly orthogonal subspace, our more flexible approach only requires that the unlearning updates are not in opposition to the retaining updates, ensuring directional coherence. This process mitigates destructive interference between competing updates and safeguards the model's performance on the retained data.

We provide theoretical guarantees that our novel forgetting loss prevents gradient explosion and ensures process stability. Furthermore, we demonstrate that our gradient projection offers first-order guarantees of a non-increase in risk for the retained tasks, formally preserving model performance. Our extensive experimental results show that FUGAS achieves thorough unlearning while simultaneously maintaining classification accuracy on retained data, showcasing its superior performance and robustness compared to state-of-the-art methods.

In summary, our work makes several key contributions. First, we propose FUGAS, a robust framework that addresses the critical challenges of instability and performance degradation with minimal training cost in federated unlearning. Second, we introduce a novel forgetting objective based on preference optimization and a flexible gradient projection mechanism, achieving effective unlearning and preserving model utility. Finally, we provide both theoretical guarantees and extensive empirical validation to demonstrate the superior performance and reliability of our method.

## 2 RELATED WORK

### 2.1 MACHINE UNLEARNING

Machine unlearning seeks to efficiently remove the influence of designated data from trained models so that the resulting model behaves as if it were trained without it (Bourtoule et al., 2021; Nguyen et al., 2022; Wang et al., 2024; Qu et al., 2023). Full retraining from scratch remains the gold standard, but its cost is prohibitive. To mitigate this, exact unlearning methods like SISA (Bourtoule et al., 2021) partition the dataset into isolated shards, enabling targeted retraining of only the affected submodels at the expense of substantial storage and recomputation (Kadhe et al., 2023; Gupta et al., 2021). Approximate unlearning methods (Marchant et al., 2022; Han et al., 2024; Foster et al., 2024; Lin et al., 2023; Wu et al., 2022b) seek greater efficiency by directly modifying the trained model's parameters to approximate a retrained state. Representative techniques include reversing the learning process through gradient ascent (Thudi et al., 2022; Chen et al., 2025; Tarun et al., 2023), leveraging influence functions to estimate data impact (Warnecke et al., 2023; Peste et al., 2021), and employing knowledge distillation to suppress targeted information (Kim et al., 2024; Zhou et al., 2025; Zhang et al., 2023b; Wang et al., 2023a; Chundawat et al., 2023). However, these methods presuppose a centralized environment with unrestricted access to data. This assumption is violated in federated learning due to its principles of data decentralization, privacy constraints, and statistical heterogeneity across clients. Consequently, the direct application of centralized unlearning techniques to the federated setting is infeasible.

## 2.2 FEDERATED UNLEARNING

Federated unlearning (Romandini et al., 2025; Liu et al., 2025a; Su & Li, 2023; Jeong et al., 2024; Liu et al., 2022) aims to erase a target client's impact from the global model while preserving utility for the remaining clients. Current approaches are commonly organized into update-storage and gradient-modification methods. Update-storage methods such as FedEraser (Liu et al., 2021) and FedRecovery (Zhang et al., 2023a) reconstruct a model without the target client by replaying or calibrating stored client updates, which incur memory that scales linearly with training rounds. Although variants attempt to reduce storage through selective retention, compression, or knowledge distillation on proxy data (Wu et al., 2022a; Yuan et al., 2023; Guo et al., 2024; Gong et al., 2022), they still rely on substantial server-side history (Cao et al., 2023; Guo et al., 2024; Wang et al., 2023b). In contrast, gradient-modification methods avoid this storage burden by formulating unlearning as an optimization problem and typically perform gradient ascent on the forgetting set to reverse prior updates (Halimi et al., 2022; Alam et al., 2024; Gong et al., 2021b). To mitigate adverse effects on model utility, some studies project ascent into a subspace that is orthogonal to the updates of retained clients (Li et al., 2023), while others introduce bounded objectives to prevent gradient explosion (Pan et al., 2025). However, gradient-modification introduce instability through gradient explosion and conflict with directions needed to maintain performance, especially under non-IID data. Our work addresses these limitations with a memory efficient federated unlearning mechanism that pairs a novel forgetting objective with a flexible projection to keep unlearning steps compatible with retained clients' gradients. The design avoids storing historical updates and yields stable progress with strong utility.

## 3 PRELIMINARY

We consider a federated learning setting with a collection of clients $k \in [K]$, each possessing a private data dataset $\mathcal{D}_k$, and a central server coordinates the training of a global model $\theta_G^*$:

$$\theta_G^* = \arg\min_\theta \sum_{k=1}^{K} p_k \mathcal{L}_k(D_k; \theta), \tag{1}$$

where $p_k$ are non-negative weights, typically proportional to the size of each client's dataset. In each federated round $t$, each client $k$ performs gradient descent on its private data to update local model $\theta_k^{t+1}$. The server then aggregates these updates to form the new global model, $\theta^{t+1} \leftarrow \sum p_k \theta_k^{t+1}$.

Federated unlearning can be viewed as the inverse process of federated learning and is typically categorized into sample, class, or client unlearning. This work focuses on client unlearning, where the goal is to remove the influence of a subset of forgetting clients $F \subseteq [K]$ from the trained model, while preserving the knowledge from the retained clients $R = [K] \setminus F$. The ideal target is the model that would have been obtained by training exclusively on the data of the retained clients:

$$\theta_R^* = \arg\min_\theta F_R(\theta) = \sum_{k \in R} p_k \, \mathcal{L}_k(D_k; \theta). \tag{2}$$

However, gradient conflict is the principal challenge in client unlearning. We define $g_k^t$ the gradient for client $k$ in round $t$, and aggregate gradient directions for the forgetting and retained sets as $g_F^t = \sum_{k \in F} p_k^F g_k^t$ and $g_R^t = \sum_{k \in R} p_k^R g_k^t$, respectively. Conflict can be quantified as follow:

$$\alpha^t = \langle g_F^t, g_R^t \rangle. \tag{3}$$

Harmful interference occurs when $\alpha^t < 0$, as the updates for the two sets are in opposing directions, leading to training instability. Non-IID setting magnifies this gradient conflict, where the local gradients for one client's local data can diverge significantly from those of other clients, leading to a larger variance in gradient directions across overall clients. This statistical dissimilarity is particularly detrimental in client unlearning, as it implies that the aggregate gradients $g_F^t$ and $g_R^t$ will aim to steer the model towards disparate optimal points. This conflict complicates the preservation of model utility for retained clients and can impede the unlearning process itself.

In this work, we aim to achieve efficient unlearning while preserving model performance on retained data, even under challenging non-IID conditions.

# 4 METHODOLOGY

Figure 1: The pipeline of the proposed FUGAS method.

## 4.1 FEDERATED UNLEARNING VIA SUBSPACE ESTIMATED DIRECTIONS

In this work, we reformulate the unlearning task as a preference optimization problem. Inspired by NPO (Zhang et al., 2024), we treat the model's original predictions on the forgetting samples as a negative reference point to be diverged from, and encourages the model's current outputs to move away from these reference outputs. This approach effectively erases the previously learned knowledge without resorting to unbounded loss maximization, naturally preventing the possibility of gradient explosion and promoting a more stable unlearning process.

Specifically, for each client $k \in F$ designated for unlearning, we capture the model's reference outputs $y_i^{bef} = f(x_i; \theta_k^t)$ at the beginning of an unlearning round $t$. During the local unlearning process, the client aims to produce new outputs $y_i = f(x_i; \theta_k^{t+1})$ that are dissimilar to $y_i^{bef}$. The forgetting objective for a client $k \in F$ is defined as:

$$\mathcal{L}_{forget}(\mathcal{D}_k; \theta^t) \triangleq \mathbb{E}_{(x,y) \in \mathcal{D}_k} \left[ \log \left( 1 + \exp \left( \frac{y^T y^{bef}}{\tau} \right) \right) \right], \tag{4}$$

where $\tau$ is a hyperparameter as temperature, controlling the sensitivity of the forgetting objective.

Although our novel objective addresses training stability, the fundamental challenge of gradient conflict remains. To resolve the issue of gradient conflict, FUGAS employs a dynamic gradient projection strategy during the server aggregation stage. Specifically, the server seeks the optimal update gradient $\tilde{g}^t$ that is nearest to the naive unlearning direction while satisfying all compatibility constraints:

$$\tilde{g}^t = \arg \min_{\tilde{g}} \frac{1}{2} \|\tilde{g} - g_F^t\|_2^2 \quad \text{s.t.} \quad \langle \tilde{g}, g_k^t \rangle \geq 0, \quad \forall k \in \mathcal{R}. \tag{5}$$

The constraint ensures the resulting update does not work against any retained client's objective , thereby preventing performance degradation.

To further improve computational efficiency, especially when the model dimension is high, we can solve its Lagrangian dual problem:

$$\mathcal{L}(v, \tilde{g}^t) = \frac{1}{2} \tilde{g}^{t\top} \tilde{g}^t - g_F^{t\top} \tilde{g}^t - v^\top g_R^t \tilde{g}^t, \tag{6}$$

where $v$ is the Lagrangian multiplier. By optimizing the quadratic programming (QP) problem, we can find the exact solution to the primal problem with significantly reduced overhead. This makes the FUGAS aggregation step practical for the large-scale federated system.

Once the optimal combined gradient $\tilde{g}^t$ is computed by solving the QP, the server performs the update to the global model. The model for the next communication round is updated $\theta_G^{t+1} \leftarrow \theta_G^t - \tilde{g}^t$. This update step integrates the forgetting and learning objectives, ensuring that the removal of one client's data does not hurt the integrity and performance of the model for the remaining clients. For more detailed information, please refer to the Appendix A.

## 4.2 ANALYSIS

**Bounded-Gradient Guarantees for Stable Unlearning.** Conventional unlearning methods that utilize gradient ascent on unbounded loss functions are susceptible to instabilities in the unlearning process. We provide a theoretical analysis to formalize the stability of our unlearning objective in the following proposition.

**Proposition 1.** *Under the Gradient Descent unlearning objective for the forgetting loss, the Euclidean norm of the gradient $||\nabla_\theta \mathcal{L}_{forget}||$ is bounded.*

*Proof.* First, we demonstrate that the loss function is lower-bounded, a necessary condition for stable optimization. Our goal is to make the current output $y$ dissimilar to the reference output $y^{bef}$, which corresponds to driving the similarity score $s = y^T y^{bef}$ towards $-\infty$. We analyze the behavior of the loss function as this objective is met:

$$\lim_{s \to -\infty} \mathcal{L} = \lim_{s \to -\infty} \log\left(1 + \exp\left(\frac{s}{\tau}\right)\right) = \log(1 + 0) = 0. \tag{7}$$

The loss function is lower-bounded by 0. A lower-bounded objective ensures that the optimization process has a stable target and does not drive the parameters into numerically unstable regions.

Second, we derive the gradient of the loss with respect to the model's output and show that its norm is strictly bounded, which in turn implies a bounded parameter gradient. Using the chain rule, the gradient with respect to the model parameters is $\nabla_\theta \mathcal{L} = (\nabla_y \mathcal{L}) \cdot (\nabla_\theta y)$.

The gradient of the loss with respect to the output $y$ is:

$$\nabla_y \mathcal{L} = \frac{\partial \mathcal{L}}{\partial s} \cdot \frac{\partial s}{\partial y} = \frac{1}{\tau} \sigma\left(\frac{y^T \cdot y^{bef}}{\tau}\right) y^{bef}. \tag{8}$$

We analyze the Euclidean norm of this gradient:

$$||\nabla_y L|| = \left|\left|\frac{1}{\tau} \sigma\left(\frac{y^T y^{bef}}{\tau}\right) y^{bef}\right|\right| = \frac{1}{\tau}\left|\sigma\left(\frac{y^T y^{bef}}{\tau}\right)\right| ||y^{bef}|| < \frac{1}{\tau}||y^{bef}||. \tag{9}$$

Since the sigmoid function is strictly bounded, such that $0 < \sigma(z) < 1$ for all $z \in \mathbb{R}$, we can establish a strict upper bound on the norm of the gradient. $y^{bef}$ is a fixed reference output computed at the beginning of the unlearning step. Therefore, its norm $||y^{bef}||$ is a finite constant. This proves that the gradient with respect to the model's output is strictly bounded.

Since the gradient with respect to the model's output $\nabla_y \mathcal{L}$ is bounded and the loss function is lower-bounded, which prevents the parameter norms $||\theta||$ from being forced to diverge while the Jacobian term $\nabla_\theta y$ also remains well-behaved during optimization. Consequently, the full parameter gradient $\nabla_\theta \mathcal{L}$ which is the product of these two terms, is also bounded. The optimization process is self-regulating: as the unlearning objective successfully met $y^T y^{bef} \to -\infty$, and the sigmoid term $\sigma(y^T y^{bef}/\tau)$ approaches 0, causing the gradient to vanish naturally. This self-regulating mechanism prevents runaway updates and eliminates the risk of gradient explosion.

**Gradient Conflict Resolution via Flexible Projection.** Instead of pursuing overly strict orthogonality, we adopt a more flexible approach. We require only that the final update direction does not conflict with the learning objectives of the retained clients, which is formalized by the constraint that the inner product between the corrected gradient and each retained gradient must be non-negative, implying an angle of no more than 90 degrees. This section provides a theoretical analysis to formalize the efficacy of this approach. To ground our analysis, we introduce the following assumptions:

**Assumption 1.** *The loss functions for the unlearning client $\mathcal{L}_F$ and the retained clients $\mathcal{L}_R$ are continuous and differentiable with respect to the model parameters $\theta$.*

**Assumption 2.** *The pre-unlearning process has converged, meaning the global model $\theta_{pre}$ is already near-optimal for the retained clients. Formally, further training on the retained data yields negligible parameter changes, which implies that the gradient for the retained task at these parameters is approximately zero: $||g_R(\theta_{pre})|| \approx 0$.*

**Proposition 2.** *Under Assumptions 1 and 2, the proposed gradient projection mechanism ensures that the unlearning update does not increase the loss for retained clients.*

*Proof.* Our goal is to analyze the performance degradation on the retained clients' task $\mathcal{T}_R$, after the model has been optimized on the unlearning client's objective $\mathcal{T}_F$. We define this degradation as the change in the retained loss:

$$\Delta\mathcal{L}_R = \mathcal{L}_R(\theta_F^*) - \mathcal{L}_R(\theta_R^*), \tag{10}$$

where $\theta_F^*$ and $\theta_R^*$ denotes the optimal parameters for the unlearning and retained tasks, respectively.

Based on Assumption 1, we can approximate the loss at $\theta_F^*$ using a first-order Taylor expansion around the retained-optimal point $\theta_R^*$ from the work (Lee et al., 2019) :

$$\mathcal{L}_R(\theta_F^*) \approx \mathcal{L}_R(\theta_R^*) + g_R(\theta_R^*)^\top \cdot (\theta_F^* - \theta_R^*). \tag{11}$$

Rearranging this equation yields an approximation for the performance degradation:

$$\Delta\mathcal{L}_R \approx g_R(\theta_R^*)^\top \cdot (\theta_F^* - \theta_R^*). \tag{12}$$

The parameter difference $(\theta_F^* - \theta_R^*)$ can be expressed by considering their evolution from the pre-unlearning model $\theta_{pre}$, after the optimization steps with learning rate $\eta$:

$$\theta_F^* - \theta_R^* = -\eta \left( g_F(\theta_{pre}) - g_R(\theta_{pre}) \right). \tag{13}$$

From Assumption 2, the initial model is near-optimal for retained clients, thus $g_R(\theta_{pre}) \approx 0$ and $\theta_{pre} \approx \theta_R^*$. This allows us to simplify the parameter difference:

$$\theta_F^* - \theta_R^* \approx -\eta \cdot g_F(\theta_R^*). \tag{14}$$

Substituting this result back into our expression for performance degradation gives:

$$\Delta\mathcal{L}_R \approx g_R(\theta_R^*)^\top \cdot (-\eta \cdot g_F(\theta_R^*)) = -\eta \cdot \langle g_R(\theta_R^*), g_F(\theta_R^*) \rangle. \tag{15}$$

To analyze this expression geometrically, we relate it to the cosine similarity $\Phi$ between the two task gradients at the retained-optimal point:

$$\Phi(\theta_R^*, T_R, T_F) = \frac{\langle g_R(\theta_R^*), g_F(\theta_R^*) \rangle}{\|g_R(\theta_R^*)\| \cdot \|g_F(\theta_R^*)\|}. \tag{16}$$

The performance degradation can now be expressed as:

$$\Delta\mathcal{L}_R \approx -\eta \cdot \|g_R(\theta_R^*)\| \cdot \|g_F(\theta_R^*)\| \cdot \Phi(\theta_R^*, T_R, T_F). \tag{17}$$

Since the term $-\eta \cdot \|g_R(\theta_R^*)\| \cdot \|g_F(\theta_R^*)\|$ is non-positive, the sign of the degradation $\Delta\mathcal{L}_R$ is determined by the sign of the cosine similarity $\Phi$. Our proposed projection mechanism directly constrains $\Phi$ by enforcing that the final update vector $\tilde{g}^t$ satisfies $\langle \tilde{g}^t, g_R^t \rangle \geq 0$ for all retained clients. This ensures that the cosine similarity is non-negative $\Phi \geq 0$. This proves that our flexible projection mechanism formally prevents performance degradation on retained clients' tasks.

## 5 EXPERIMENT

### 5.1 EXPERIMENT SETUP

We benchmark our approach against several representative federated unlearning methods. Retraining serves as the gold standard, where the global model is retrained from scratch on the remaining clients during the unlearning phase. FedEraser (Liu et al., 2021) reconstructs the unlearned model by leveraging historical parameter updates from the server's storage. Projected Gradient Descent (PGD) (Halimi et al., 2022) maximizes the loss on the target client's data while keeping the model parameters within an $\ell_2$-norm ball of a reference model. FedBU (Alam et al., 2024) optimizes a weighted objective that suppresses target knowledge while preserving important parameters for accuracy on retain clients. FedOSD (Pan et al., 2025) computes an orthogonal steepest descent direction that aligns with the target update while remaining orthogonal to the retained clients' gradients, couples this with an unlearning cross entropy loss at the target.

We evaluate our proposed method and all baselines on CIFAR-10 and CIFAR-100 using the same four layer CNN for image classification. Our federated learning environment consists of ten clients, with one or two client chosen uniformly at random for unlearning in each experimental run. All clients participate in each communication round. For evaluation, we assess the test accuracy on a held-out test set from the remaining clients' data distribution. Additionally, we employ backdoor attacks to further valid unlearning effectiveness (Gu et al., 2017; Bagdasaryan et al., 2020; Xie et al., 2019), which is quantified by the attacks success rates (ASR) after the unlearning process is complete. All results are averaged over three random seeds.

Table 1: Comparison of ASR and classification accuracy (mean and std.) for various methods on CIFAR-10 and CIFAR-100. 'uc' indicates the number of unlearning clients.

| Algorithm | CIFAR-10 Non-IID | | | | CIFAR-10 IID | | | | CIFAR-100 Non-IID | | | | CIFAR-100 IID | | | |
|---|---|---|---|---|---|---|---|---|---|---|---|---|---|---|---|---|
| | uc = 1 | | uc = 2 | | uc = 1 | | uc = 2 | | uc = 1 | | uc = 2 | | uc = 1 | | uc = 2 | |
| | ASR | Accuracy | ASR | Accuracy | ASR | Accuracy | ASR | Accuracy | ASR | Accuracy | ASR | Accuracy | ASR | Accuracy | ASR | Accuracy |
| $w_0$ | .914 | .467(.012) | .996 | .257(.005) | .670 | .539(.016) | .929 | .345(.003) | .890 | .250(.055) | .939 | .233(.005) | .785 | .258(.065) | .886 | .254(.013) |
| Retrain | .032 | .576(.004) | .062 | .578(.002) | .023 | .700(.003) | .021 | .691(.001) | .001 | .298(.002) | .002 | .298(.000) | .005 | .325(.002) | .005 | .320(.002) |
| Gradient ascent | .000 | .111(.000) | .000 | .010(.000) | .000 | .099(.002) | .000 | .099(.001) | .000 | .010(.000) | .000 | .011(.000) | .000 | .013(.000) | .000 | .013(.000) |
| PGD | .000 | .108(.004) | .000 | .108(.003) | .000 | .099(.003) | .000 | .103(.000) | .000 | .029(.016) | .000 | .011(.001) | .000 | .010(.001) | .000 | .011(.000) |
| FedBU | .000 | .049(.012) | .000 | .064(.001) | .000 | .356(.014) | .000 | .336(.039) | .000 | .116(.009) | .000 | .056(.010) | .000 | .132(.025) | .000 | .091(.004) |
| FedEraser | .019 | .353(.009) | .097 | .371(.014) | .047 | .548(.030) | .051 | .572(.005) | .005 | .164(.015) | .001 | .176(.002) | .004 | .202(.014) | .003 | .224(.014) |
| FedOSD | .005 | .482(.016) | .006 | .299(.017) | .000 | .545(.014) | .000 | .562(.009) | .001 | .281(.007) | .003 | .295(.002) | .001 | .312(.007) | .000 | .320(.005) |
| FUGAS | .010 | **.533(.009)** | .013 | .462(.010) | .006 | **.686(.017)** | .008 | **.674(.007)** | .006 | **.295(.001)** | .003 | **.299(.001)** | .009 | **.325(.002)** | .008 | **.324(.002)** |

## 5.2 RESULTS

We report the performance of various federated unlearning methods in terms of ASR and classification accuracy under both IID and Non-IID settings in Table 1. We use the MIA metric to further compare our method with other baselines in Appendix C.2.

As presented in Table 1, foundational approaches like Gradient Ascent and its constrained variant PGD successfully reduce the ASR to zero, but they suffer from catastrophic forgetting, causing the model's accuracy to collapse to near-random levels. This outcome illustrates the instability of maximizing an unbounded loss function. Similarly, aggressive unlearning baselines such as PGD and FedBU effectively nullify the ASR but at the cost of catastrophic forgetting, leading to a collapse in model accuracy. While methods like FedEraser and FedOSD achieve a better balance by reducing the ASR while retaining some model utility. However, their performance reveals a clear trade-off, as they still incur a significant drop in accuracy compared to the retraining baseline. The retraining approach serves as the gold standard, successfully restoring high accuracy while maintaining a negligible ASR, but it incurs prohibitive computational costs.

In contrast to the baselines, our method FUGAS balances unlearning efficacy with model utility preservation across all evaluated settings. FUGAS consistently reduces the ASR to near-zero levels, indicating a thorough erasure of the targeted client's data. Most importantly, FUGAS excels in maintaining high classification accuracy, outperforming all other unlearning methods and closely matching the strong performance of the computationally intensive Retrain approach. This success is directly attributable to our core methodology. The novel preference optimization objective effectively erases knowledge by encouraging divergence from reference outputs, which circumvents the gradient explosion problem. Furthermore, the flexible gradient projection strategy at the server level is crucial for preserving utility, resolving gradient conflicts and ensuring the global update is not detrimental to any remaining client.

## 5.3 ABLATION STUDY

Table 2: Ablation with gradient constraint under different positive and negative sample selection strategies. '-' means negative sample and '+' means positive sample.

| | CIFAR-10 Non-IID | | CIFAR-10 IID | | CIFAR-100 Non-IID | | CIFAR-100 IID | |
|---|---|---|---|---|---|---|---|---|
| | ASR | Accuracy | ASR | Accuracy | ASR | Accuracy | ASR | Accuracy |
| Uniform Distribution$^+$ | .012 | .505(.016) | .026 | .673(.008) | .010 | .292(.001) | .010 | .317(.004) |
| Data Augmentation$^+$ | .020 | .525(.005) | .030 | **.693(.006)** | .008 | .273(.026) | .016 | .321(.004) |
| Label$^-$ | .043 | .503(.025) | .003 | .681(.013) | .006 | .281(.017) | .011 | .323(.004) |
| FUGAS | .010 | **.533(.009)** | .006 | .686(.017) | .006 | **.295(.001)** | .009 | **.325(.002)** |

**The Effectiveness of Sample Selection Strategies.** We evaluate the effective of positive and negative preference samples in the unlearning objective. We introduce additional positive samples through sampling from a uniform distribution (Uniform Distribution$^+$) and applying data augmentation (Data Augmentation$^+$). Furthermore, we use the negative preference with ground-truth labels as the negative reference (Label$^-$) for comparison. The results are summarized in Table 2.

Specifically, Label$^-$ proves overly aggressive while it achieves a notably low ASR, it leads to a consistent degradation in classification accuracy, particularly in the Non-IID setting. Conversely,

while introducing explicit positive samples can maintain or slightly improve final model accuracy, these variants consistently yield a higher residual ASR. We attribute this to the fact that providing a less-defined or overly broad positive target can dilute the unlearning signal, allowing backdoor knowledge to persist. Moreover, incorporating additional positive samples incurs extra training costs. In contrast, our approach using the pre-unlearning model's outputs as a negative reference provides a contextually relevant target at low computational cost, effectively guiding the model to forget backdoor influences.

Table 3: Ablation of gradient conflict mitigation strategies.

|  | CIFAR-10 Non-IID | | CIFAR-10 IID | | CIFAR-100 Non-IID | | CIFAR-100 IID | |
|---|---|---|---|---|---|---|---|---|
|  | ASR | Accuracy | ASR | Accuracy | ASR | Accuracy | ASR | Accuracy |
| Naive Unlearning | .000 | .111(.000) | .000 | .099(.002) | .000 | .010(.000) | .000 | .013(.000) |
| FUGAS w/o Constraint | .005 | .428(.022) | .000 | .575(.041) | .001 | .285(.006) | .002 | .320(.003) |
| FUGAS w/ Orthogonal | .019 | .466(.040) | .003 | .495(.081) | .004 | .273(.006) | .006 | .321(.007) |
| FUGAS | .010 | **.533(.009)** | .006 | **.686(.017)** | .006 | **.295(.001)** | .009 | **.325(.002)** |

**The Effectiveness of Angle Constraint.** We perform an ablation study to quantify the effect of imposing an angle constraint on the gradients of unlearned and retained clients. We first establish a baseline with Naive Unlearning, which applies a standard cross-entropy unlearning objective without any constraints, thereby revealing the unmodified effects of gradient conflict. We then analyze two variations of our proposed method: FUGAS without any constraint and FUGAS with a strict orthogonality constraint. The result is presented in the Table 3.

Naive Unlearning baseline successfully reduces the ASR to zero but suffers from catastrophic forgetting, causing accuracy to collapse to near-random levels. This collapse is a direct result of destructive interference from unconstrained gradient conflicts. While our bounded unlearning objective alone (FUGAS w/o Constraint) substantially mitigates this issue and preserves significant model utility, it remains suboptimal. Conversely, enforcing strict orthogonality (FUGAS w/ Orthogonal) degrades performance compared to the unconstrained version in several cases, as it excessively restricts the unlearning updates and discards potentially useful information. In contrast, our full FUGAS method consistently achieves the highest accuracy, which demonstrates that by flexibly ensuring unlearning updates are merely compatible with retaining updates rather than strictly orthogonal. Therefore, our acute angle constraint strikes a more effective balance, successfully resolving conflicts while preserving the performance of the global model.

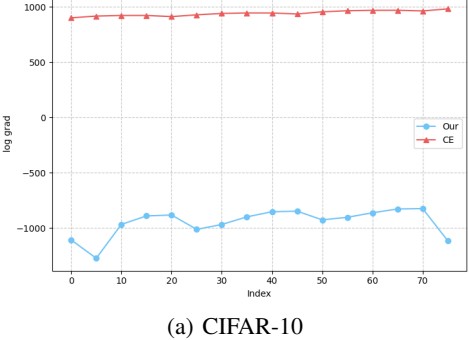
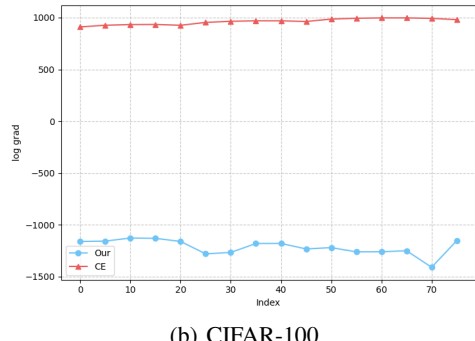

(a) CIFAR-10                                       (b) CIFAR-100

Figure 2: Comparison of gradient magnitudes for our objective versus a cross-entropy maximization objective on (a) CIFAR-10 and (b) CIFAR-100. The y-axis plots the log of the gradient norm.

## 5.4 ANALYSIS

**Analysis of Gradient Stability.** To analyze the instability inherent in naive unlearning and to highlight the stability offered by our proposed objective, we compare the gradient magnitudes of our method against a standard cross-entropy (CE) maximization baseline, as illustrated in Figure 2.

The gradient norms associated with the CE objective are consistently several orders of magnitude larger than those produced by our method. This gradient explosion phenomenon provides a direct explanation for the catastrophic forgetting observed in Table 1. The explosive gradients destructively interfere with the learned representations, effectively nullifying the model's utility. In contrast, our bounded unlearning objective maintains small and stable gradient magnitudes throughout the unlearning process. Even without the final gradient projection step, our bounded objective alone is sufficient to prevent catastrophic forgetting, retaining a high level of accuracy as shown in Table 3. This stable foundation ensures that the unlearning process can precisely remove specific knowledge without hurting the model's performance on the retained data, enabling a favorable balance between unlearning efficacy and utility.

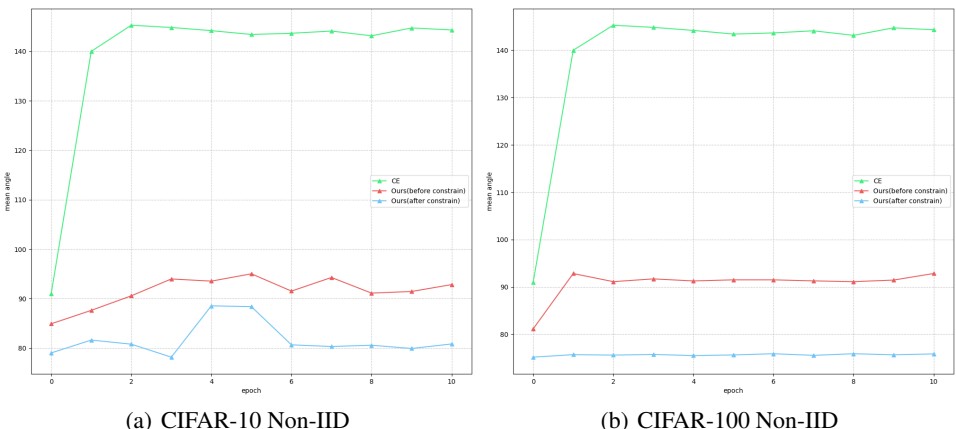

(a) CIFAR-10 Non-IID        (b) CIFAR-100 Non-IID

Figure 3: Visualization of our gradient projection mechanism in mitigating gradient conflict.

**Analysis of Gradient Harmony.** We empirically investigate the critical issue of gradient conflict during the global aggregation phase of federated unlearning, a phenomenon visualized in Figure 3. This conflict arises when the gradient updates from the unlearning client are misaligned with the aggregate gradient of the retaining clients, leading to destructive interference. Our analysis shows that employing a CE loss for unlearning consistently results in an obtuse angle between the unlearning and retaining gradients, visually confirming a state of severe conflict. This direct opposition explains the performance collapse observed in Table 1, where the model's accuracy on retained data degrades catastrophically despite a successful reduction in ASR.

In contrast, our unlearning objective alone significantly alleviates this issue by inherently stabilizing the gradients and reducing the angle of opposition. This effect is further enhanced by our flexible acute angle projection mechanism, which explicitly enforces gradient compatibility by projecting the unlearning update to ensure it is not in opposition to the retaining updates. This resolution of gradient conflict is the key mechanism that enables our method to achieve effective unlearning while simultaneously safeguarding high performance on the remaining clients task.

## 6 CONCLUSION

In this work, we introduced Federated Unlearning with GrAdient Shielding (FUGAS), a framework designed to address stability and performance degradation challenges in federated unlearning. By reformulating unlearning as preference optimization against a negative reference, FUGAS replaces unbounded loss maximization with a bounded objective that stabilizes training. The flexible projection of unlearning updates reduces gradient conflict and preserves utility, ensuring that unlearning updates do not destructively interfere with the knowledge learned from the retained data. We prove the stability of our unlearning process and the preservation of model performance on retained data. Extensive experiments show that FUGAS achieves thorough removal of targeted information while matching or improving accuracy on retained data. FUGAS requires no storage of historical updates, significantly reducing memory overhead. Future work includes extensions to personalized federated learning and foundation models, as well as more comprehensive privacy and security evaluations.

## ETHICS STATEMENT

This paper presents work whose goal is to advance the field of Federated Unlearning privacy and user autonomy by enabling data contributors to revoke their influence on trained models, supporting the "right to be forgotten". The authors have read and comply with the ICLR Code of Ethics. The research did not involve human subjects, animal experiments, or personally identifiable data. All experiments were conducted on publicly available benchmarks and open-source models. We have carefully considered the broader impacts and believe that this work poses no foreseeable risks of harm while contributing to the development of robust and secure models.

## REPRODUCIBILITY STATEMENT

We are committed to ensuring the reproducibility of our research. Our complete implementation, including the source code to replicate all experiments and generate the figures and tables presented in this paper, will be made publicly available upon publication. A detailed description of our proposed method, Federated Unlearning with GrAdient Shielding (FUGAS), is provided in Section 4, with a step-by-step pseudocode available in the Appendix **??**. The theoretical claims regarding the stability of our unlearning objective and the performance guarantees of our gradient projection mechanism are formally stated and proven in Section 4.2, with additional mathematical derivations provided in Appendix A. All experimental details, including the datasets (CIFAR-10 and CIFAR-100), the non-IID data partitioning strategy using a Dirichlet distribution, the specific model architecture, and a comprehensive list of all hyperparameters (such as learning rates, batch size, and the number of communication rounds for each phase), are documented in Section and further expanded in Appendix C.1 to facilitate the exact replication of our results.

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

## A   METHODOLOGICAL DETAILS AND DERIVATIONS

We take the derivative with respect to $\tilde{g}^t$ in Eq. 6:

$$\nabla_{\tilde{g}^t} \mathcal{L} = \tilde{g}^t + g_F^t - g_R^{t\top} v = 0. \tag{18}$$

Solving gives:

$$\tilde{g}^t = g_R^{t\top} v + g_F. \tag{19}$$

Substituting $\tilde{g}^t$ back into the Lagrangian yields the dual problem:

$$
\begin{aligned}
\underset{v}{\text{minimize}} \quad \mathcal{L}(v) \quad &= \frac{1}{2} \left( g_R^{t\top} v + g_F \right)^\top \left( g_R^{t\top} v + g_F \right) \\
&\quad - g_F^\top \left( g_R^{t\top} v + g_F \right) \\
&\quad - v^\top g_R^t \left( g_R^{t\top} v + g_F \right) \\
&= -\frac{1}{2} v^\top g_R^t g_R^{t\top} v - v^\top g_R^t g_F \\
\text{subject to} \quad &\quad v \geq 0.
\end{aligned}
\tag{20}
$$

Here, $v \in \mathcal{R}^{c-1}$ is the Lagrange multiplier. By optimizing the dual function, we reduce the problem from optimizing $\tilde{g}^t \in \mathcal{R}^p$ to optimizing $v \in \mathcal{R}^{c-1}$, simplifying the problem and significantly reducing computation time.

---

**Algorithm 1** FUGAS

---

**Require:** Pretrained model $\theta_G^0$, unlearning round $T$, client set $K$  unlearning client set $F$;
 1: Client set $F$ request for unlearning
 2: **for** $t = 0$ **to** $T - 1$ **do**
 3:     Server send global model $\theta_G^t$ to all client $i \in K$
 4:     **if** client $i \in F$ **then**
 5:         $\theta_i^t \leftarrow$ Client $i$ performs unlearning using the loss $\mathcal{L}$
 6:         Client $i$ upload $g_i^t = (\theta_G^t - \theta_i^t)$
 7:     **else**
 8:         $\theta_i^t \leftarrow$ Each client $i$ performs local training
 9:         Client $i$ upload $g_i^t = (\theta_G^t - \theta_i^t)$
10:     **end if**
11:     $g_i^t \leftarrow \frac{n_i}{\sum_{j=1}^k n_j} \cdot g_i^t, \forall i \in K, i \notin F$
12:     $g_F^t \leftarrow \text{sum}(\cdots, g_i^t, \cdots)/F, \forall i \in F$
13:     Calculate final unlearning gradient $\tilde{g}^t$ by Eq. 20
14:     $\theta_G^{t+1} \leftarrow \theta_G^t - \tilde{g}^t$
15: **end for**

---

## B   PIPELINE

Federated Unlearning (FUGAS) consists of several stages, starting from pretraining and continuing through the unlearning phase, followed by the global model update. In the pretraining phase, all clients collaboratively train a global model. Each client performs local updates based on its own data, and the server aggregates these updates to form a global model that generalizes well across diverse data distributions.

Once the pretraining phase is complete, the unlearning phase begins. In this phase, a client that requests unlearning aims to effectively forget its contribution to the global model while ensuring the model's utility for other clients. Client $k \in F$ saves the output $y^{bef}$ on its dataset at the current

model state as negative samples for unlearning. Client $u$ performs local unlearning by optimizing a specific loss function $\mathcal{L}$. While all other clients continue their standard learning operations, updating the global model based on their local data.

During the communication phase, both unlearning clients and remaining clients send their gradient updates to the server. In the global aggregation phase, the server collects the gradient updates from all clients. The server synthesizes the gradients $g_i^t$ from all remaining clients into a matrix $g_F^t$. When there are multiple unlearning clients, we average the gradient updates returned by multiple clients to obtain $g_F^t$. To alleviate gradient conflicts between forgotten clients and remaining clients, we solve a constrained optimization problem to obtain the final gradient vector $\tilde{g}^t$. This constraint ensures that the dot product between the final gradient $\tilde{g}^t$ and the gradient direction of the retained clients is positive, thereby preventing catastrophic forgetting of model performance on remaining clients.

Finally, the server applies the projected gradient to update the global model $\theta_G^{t+1}$. The updated global model is then broadcast to all clients, ensuring that the global model remains consistent across the federation. This process repeats iteratively until the unlearning objective is met. The pseudo-code of FUGAS is shown in Algorithm 1.

## C  EXPERIMENT

### C.1  EXPERIMENT SETUP

**Datasets.** To simulate diverse federated learning environments, we consider both IID and non-IID data distributions across clients. In the IID setting, the entire data is shuffled and partitioned uniformly, ensuring each client receives an equal number of samples with a balanced class distribution. In the non-IID setting, client specific class proportions are drawn from a Dirichlet distribution with a concentration parameter $\alpha = 0.1$. Furthermore, to emulate a practical scenario where evaluation is performed locally, we ensure that each client's test data distribution is consistent with its local training data distribution.

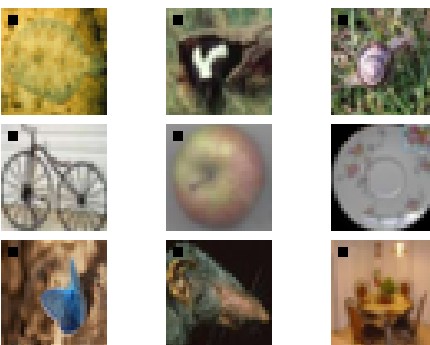

Figure 4: Examples of backdoor trigger pictures.

**Implement Details.** We implement all methods beginning with 500 pre-training rounds to establish a stable global model, with local optimization performed on each client using SGD with a batch size of 32 and one local epoch per round. Following pre-training, we execute a 10-round unlearning phase, where the designated unlearning protocol is applied. This is succeeded by a 10-round post-training phase involving only the remaining clients to recover model utility. In the training phase, we use a learning rate of 0.001, and in the forgetting phase, we use a learning rate of 1e-4. For each algorithm, we select three random seeds for experimentation and calculated the mean and variance of the results.

We evaluate the robustness of our approach through the execution of Membership Inference Attacks and backdoor attacks. MIA assesses the model's ability to protect the privacy of the unlearning clients' data, determining whether an attacker can infer whether a particular sample was included in the training data of the model (Shokri et al., 2017; Bai et al., 2024). A lower MIA accuracy indicates stronger privacy protection. On the other hand, backdoor attacks are used to test the model's resilience to malicious alterations during the unlearning process, where an attacker attempts to insert

Table 4: MIA precision and classification accuracy comparison pre-unlearning under CIFAR-10 and CIFAR-100 using different algorithms. 'uc' indicates the number of unlearning clients.

| Algorithm | CIFAR-10 Non-IID | | | | CIFAR-10 IID | | | | CIFAR-100 Non-IID | | | | CIFAR-100 IID | | | |
|---|---|---|---|---|---|---|---|---|---|---|---|---|---|---|---|---|
| | uc = 1 | | uc = 2 | | uc = 1 | | uc = 2 | | uc = 1 | | uc = 2 | | uc = 1 | | uc = 2 | |
| | MIA Acc | Accuracy | MIA Acc | Accuracy | MIA Acc | Accuracy | MIA Acc | Accuracy | MIA Acc | Accuracy | MIA Acc | Accuracy | MIA Acc | Accuracy | MIA Acc | Accuracy |
| $w_0$ | .979 | .578(.002) | .979 | .578(.002) | .975 | .707(.000) | .975 | .707(.000) | .971 | .297(.002) | .971 | .297(.002) | .984 | .330(.007) | .984 | .330(.007) |
| Retrain | .748 | .573(.002) | .751 | .571(.001) | .748 | .704(.004) | .756 | .698(.003) | .749 | .298(.002) | .750 | .297(.002) | .752 | .327(.006) | .756 | .322(.006) |
| PGD | .746 | .142(.009) | .756 | .132(.000) | .755 | .279(.103) | .749 | .102(.022) | .750 | .194(.054) | .748 | .045(.045) | .795 | .287(.022) | .796 | .304(.034) |
| FedEraser | .717 | .334(.009) | .715 | .344(.012) | .762 | .559(.021) | .773 | .558(.021) | .771 | 163(.004) | .756 | .177(.004) | .753 | .228(.032) | .755 | .229(.033) |
| FedOSD | .759 | .448(.062) | .745 | .517(.007) | .757 | .373(.192) | .752 | .230(.155) | .752 | .262(.030) | .753 | .268(.021) | .809 | .245(.094) | .798 | .296(.032) |
| FUGAS | .705 | **.564(.007)** | .708 | **.591(.004)** | .748 | **.691(.022)** | .755 | **.688(.010)** | .747 | **.294(.006)** | .748 | **.299(.003)** | .801 | **.332(.006)** | .816 | **.330(.002)** |

Table 5: Comparison of ASR and classification accuracy (mean and std.) for various methods on CIFAR-10 and CIFAR-100 after post-training. 'uc' indicates the number of unlearning clients.

| Algorithm | CIFAR-10 Non-IID | | | | CIFAR-10 IID | | | | CIFAR-100 Non-IID | | | | CIFAR-100 IID | | | |
|---|---|---|---|---|---|---|---|---|---|---|---|---|---|---|---|---|
| | uc = 1 | | uc = 2 | | uc = 1 | | uc = 2 | | uc = 1 | | uc = 2 | | uc = 1 | | uc = 2 | |
| | ASR | Accuracy | ASR | Accuracy | ASR | Accuracy | ASR | Accuracy | ASR | Accuracy | ASR | Accuracy | ASR | Accuracy | ASR | Accuracy |
| $w_0$ | .914 | .467(.012) | .996 | .257(.005) | .670 | .539(.016) | .929 | .345(.003) | .890 | .250(.055) | .939 | .233(.005) | .785 | .258(.065) | .886 | .254(.013) |
| Retrain | .032 | .576(.004) | .062 | .578(.002) | .023 | .700(.003) | .021 | .691(.001) | .001 | .298(.002) | .002 | .298(.000) | .005 | .325(.002) | .005 | .320(.002) |
| FedEraser | .014 | .373(.011) | .075 | .395(.006) | .043 | .585(.024) | .030 | .607(.010) | .004 | .184(.017) | .001 | .202(.001) | .004 | .233(.027) | .017 | .246(.032) |
| FedOSD | .068 | .546(.002) | .135 | .515(.005) | .074 | .703(.003) | .065 | .696(.003) | .040 | .295(.001) | .015 | .299(.002) | .029 | .323(.001) | .023 | .323(.002) |
| FUGAS | .067 | .545(.004) | .168 | .527(.002) | .069 | .703(.002) | .042 | .696(.003) | .037 | .295(.001) | .015 | .299(.001) | .027 | .325(.003) | .018 | .323(.002) |

a trigger into the model that causes it to behave maliciously under specific conditions. ASR is used to measure the success of the attack. Example images of backdoor attacks can be found in Figure 4, illustrating how an attacker can manipulate the model's behavior by introducing malicious triggers. By convention, higher is better for forget accuracy, retain accuracy, and test accuracy, while lower is better for MIA attack accuracy and cost time. In the complexity analysis we invert MIA attack rate and when plotting.

To be continued, all experiments are conducted on a machine with an Intel(R) Xeon(R) Gold 6348 CPU @ 2.60GHz and an A100 GPU. The setup and parameters used in this experiment part are consistent with those detailed in Section 6.

## C.2 RESULT

In the experiments whose result is Table 4, we evaluate the performance of our FUGAS method under MIA attacks using CIFAR-10 and CIFAR-100 datasets, considering both IID and Non-IID settings. We explore scenarios where the unlearning clients consist of either one or more clients, simulating different unlearning configurations. As shown in Table 4, FUGAS consistently demonstrates superior robustness against MIA attacks compared to other methods like FedEraser and FedOSD. Specifically, FUGAS achieves low MIA accuracy while maintaining strong classification performance across different experimental configurations. The consistent performance across different scenarios demonstrates the robustness of our approach in ensuring both privacy and model accuracy, confirming the effectiveness of FUGAS in federated unlearning tasks.

In the experiment result Table 5, we focus on the unlearning phase and the post-training process, which is complementary to the unlearning task. The post-training phase aims to recover the model's utility after the unlearning operation, ensuring that the model's performance remains intact while the influence of unlearning clients is effectively removed. Our method incorporates a gradient projection strategy as the post-training approach used in FedOSD (Pan et al., 2025). In the post-training phase, each retained client computes its gradient, and if the gradient is aligned with the unlearning direction, it is projected onto a subspace that avoids reintroducing knowledge from the forgotten clients. The global model is then updated based on the aggregated projections of the gradients from the retained clients. This method ensures that the model does not revert to its original state after unlearning, maintaining the effectiveness of the forget operation.

As shown in Table 5, the results indicate that FUGAS performs comparably to FedOSD in the post-training phase, demonstrating its robustness. In all of the scenerios, FUGAS achieves good classification accuracy and low ASR after the post-training phase. These results confirm that our method effectively integrates unlearning with post-training to preserve model performance while preventing the model from regaining knowledge of forgotten clients. Based on it, our potential future direction is to explore additional post-training methods and their combination with FUGAS to further enhance the model's robustness while ensuring that the unlearning operation is not undone.

Table 6: "F-Acc" is the accuracy of the knowledge unique to the unlearning client, "C-Acc" is for overlapping knowledge, and "R Acc" is for the knowledge unique to the remaining clients

| | CIFAR-10 | | | CIFAR-100 | | |
|---|---|---|---|---|---|---|
| | F-Acc | C-Acc | R-Acc | F-Acc | C-Acc | R-Acc |
| Retrain | .000(.000) | **.358(.008)** | **.704(.005)** | .000(.000) | **.209(.012)** | **.333(.001)** |
| FedEraser | .000(.000) | .000(.000) | .601(.003) | .000(.000) | .000(.000) | .257(.003) |
| FedOSD | .001(.000) | .001(.001) | .679(.019) | .015(.006) | .047(.003) | .321(.002) |
| FUGAS | .008(.002) | .177(.008) | .699(.005) | .026(.005) | .192(.049) | .332(.002) |

## C.3 KNOWLEDGE INTERFERENCE

To investigate the maintenance of overlapping knowledge among clients, we partition the CIFAR-10 and CIFAR-100 datasets as follows: 90% of the data labeled as 0 and all data labeled as 1 are assigned to a specific client, while the remaining data are randomly distributed among other clients. After applying the forgetting mechanism, we evaluate the classification accuracy for three types of knowledge: 1) the unique knowledge of the forgetting client (data labeled as 1); 2) the overlapping knowledge (the dataset labeled as 0 in remaining clients); and 3) the unique knowledge of the remaining clients (all data excluding labels 0 and 1).

As shown in Tab 6, all algorithms demonstrate substantial forgetting of the unique knowledge held by the unlearning client, with F-Acc approaching zero, indicating effective removal of client-specific information. In contrast, for the overlapping knowledge shared across clients (C-Acc), the Retrain baseline maintains high retention rates, while other methods like FedEraser and FedOSD suffer from significant degradation, dropping to near-zero performance. Our proposed FUGAS method strikes a favorable balance, thereby effectively safeguarding shared knowledge without compromising the unlearning objective. Regarding the unique knowledge of the remaining clients (R-Acc), FUGAS preserves relatively high accuracy, closely matching the Retrain baseline. Overall, these results validate FUGAS's balanced trade-off between targeted unlearning and global model utility, addressing key limitations in prior frameworks.

## C.4 COMPLEXITY ANALYSIS

The evaluation metrics in this section includes test accuracy, forget accuracy, retain accuracy, MIA attack, and cost time, are visualized through Figure 5, which provides a comprehensive overview of the model's performance compared to baseline algorithms. Specifically, Forget accuracy measures the accuracy of the unlearning clients on the last epoch of the train phase during unlearning, indicating how well the model forgets the data of the targeted clients; Retain accuracy reflects the accuracy of the remaining clients on the last epoch of the train phase during unlearning, showing how well the model retains the knowledge from the clients that were not selected for unlearning; Test accuracy represents the accuracy of the remaining clients on the last epoch of the test phase during unlearning, indicating the overall generalization ability of the model after unlearning; MIA attack measures the accuracy of the Membership Inference Attack, assessing how well the model prevents attackers from inferring if a client's data was included in the training set; Cost time indicates the total computational time spent in the unlearning process, providing insight into the efficiency of the method. Because Retrain incurs much longer running cost time, we exclude it from the cost time axis.

The Retrain method serves as an upper bound, showing the best performance. However, while Retrain achieves the highest performance, it comes at the cost of significantly higher computational time, making it impractical for large-scale applications. In contrast, our method, FUGAS, efficiently achieves effective unlearning while maintaining good performance on the retained clients, all while keeping the time cost manageable. The result is in Figure 5

When comparing FUGAS with other methods like FedOSD (Pan et al., 2025) and PGD (Halimi et al., 2022) , we observe that FUGAS strikes a better balance between privacy protection and model utility. In terms of MIA attack resistance, FUGAS performs better than FedOSD and PGD. Moreover, FUGAS excels in maintaining high retain accuracy, ensuring that the model retains most of

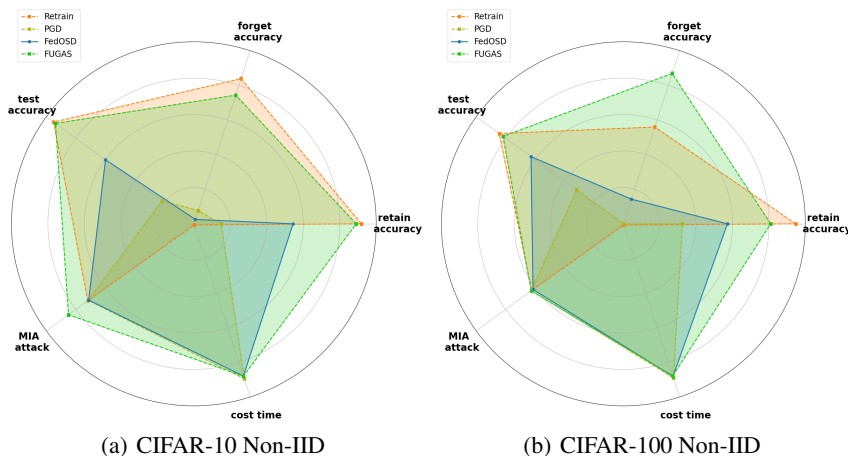

(a) CIFAR-10 Non-IID        (b) CIFAR-100 Non-IID

Figure 5: Performance comparison of various indicators between FUGAS and three other baselines under MIA attack.

its utility even after unlearning certain clients' data. As shown in the result plots, FUGAS consistently outperforms FedOSD and PGD in multiple aspects, particularly in forget accuracy, while also remaining more computationally efficient than Retrain.

Turning to the method-wise comparison, PGD sacrifices utility on both the forgotten and retained domains, yielding smaller polygons despite decent privacy, which is evidence of over-aggressive updates. FedOSD improves retain/test accuracy over PGD and offers competitive privacy, but its utility on the forgotten domain remains limited, suggesting residual gradient interference under Non-IID heterogeneity. Across both datasets, FUGAS expands the polygon along all four effectiveness axes (forget accuracy, retain accuracy, test accuracy, and declien of MIA) while matching peers on time, thereby maximizing the overall area. Notably, FUGAS closes much of the gap to Retrain on the accuracy axes, approaching the upper bound without the prohibitive runtime. This aligns with our design: per-client update correction mitigates destructive interference, removing client-specific signals (lower MIA) while preserving task-relevant structure (higher accuracies). Overall, FUGAS achieves the most favorable balance of privacy, utility, and efficiency among practical methods, making it a strong choice for real-world federated unlearning.

## D  THE USE OF LARGE LANGUAGE MODELS (LLMS)

In the preparation of this manuscript, we utilized a Large Language Model (LLM) as a writing assistant. The use of the LLM was strictly limited to improving the language and readability of the text. This included tasks such as correcting grammar and spelling, rephrasing sentences for clarity and flow, and ensuring stylistic consistency.

The LLM did not contribute to the research ideation, the development of the FUGAS methodology, the theoretical analysis, the design of the experiments, or the interpretation of the results. All core scientific concepts, mathematical formulations, experimental findings, and conclusions presented in this paper are the original work of the human authors. The authors have carefully reviewed and edited all LLM-assisted text to ensure its technical accuracy and take full responsibility for the entire content of this paper.

