# OpenReview forum: "Federated Unlearning with Gradient Shielding"
_ICLR.cc/2026/Conference — ICLR 2026 Conference Withdrawn Submission_

### Official Review · Reviewer_wJMZ · 2025-10-27

**Soundness:** 3
**Presentation:** 3
**Contribution:** 3
**Rating:** 4
**Confidence:** 5

**Summary:**

This paper presents a novel and well-designed framework, Federated Unlearning with GrAdient Shielding (FUGAS), to address critical challenges in federated unlearning. The core contributions—a bounded forgetting loss based on preference optimization and a flexible gradient projection mechanism—are innovative and effectively tackle the fundamental problems of gradient explosion and conflicting updates between forgetting and retaining clients. The methodology is sound and clearly articulated. The experimental evaluation is exceptionally thorough, covering diverse non-IID settings, strong baselines, and a comprehensive set of metrics including accuracy, attack success rate (ASR), and membership inference attacks (MIA). The results strongly support the paper's claims, demonstrating that FUGAS achieves a superior balance between effective unlearning and utility preservation for remaining clients, without the high memory overhead of previous methods. Overall, this is a high-quality paper that makes a significant and practical contribution to the federated learning community and is well-suited for publication.

**Strengths:**

The proposed FUGAS framework is innovative, particularly its use of a preference optimization-based objective for a bounded unlearning loss. This is an elegant solution to the gradient explosion problem that plagues methods based on simple loss maximization.
The method is memory-efficient as it does not require storing historical client updates, which is a major bottleneck for many federated unlearning approaches. This makes the approach highly practical for deployment in large-scale, real-world systems.
The paper's claims are well-supported by a combination of theoretical analysis and thorough empirical evaluations. The experiments cover multiple datasets, IID and non-IID settings, and a comprehensive set of metrics, demonstrating a clear and consistent advantage over strong baselines.

The flexible gradient projection mechanism is a more nuanced approach to mitigating gradient conflicts than prior work that enforces strict orthogonality. By only requiring directional compatibility, it preserves more information from the unlearning update, leading to better model utility as shown in the ablation studies.

**Weaknesses:**

1. The theory lacks formal guarantees on how closely the resulting model approximates the gold-standard retrained model. The paper should be strengthened by adding an analysis that bounds the divergence between the unlearned and retrained models.

2. The theoretical analysis for utility preservation hinges on a strong assumption that the model has already converged for retained clients. The authors should provide an analysis or empirical study on the method's performance when this assumption is violated, which is common in practice.

3. The paper does not analyze the computational scalability of the gradient projection step, which involves solving a constrained optimization problem. An analysis of how the aggregation time scales with the number of retained clients is needed to assess its practicality in large-scale federations.

4. The empirical evaluation is limited to a simple CNN architecture on relatively small-scale image classification datasets. To demonstrate broader applicability, experiments should be extended to more complex models like ResNet and diverse data modalities.

5. The impact of the temperature hyperparameter T in the forgetting loss function is not investigated. A sensitivity analysis should be included to show how T affects the balance between unlearning effectiveness and model utility.

6. The effectiveness of the proposed method relies on the quality of the pre-unlearning model's predictions, which may be poorly calibrated. The analysis would be more convincing with a discussion on how the initial model's performance on the forget set impacts the unlearning process.

7. The experimental results for non-IID settings, while strong, could be further challenged with more extreme data heterogeneity. The paper should consider evaluating under more severe non-IID partitions to better test the robustness of the gradient shielding mechanism.

**Questions:**

1. Regarding the theoretical contribution, could the authors provide any analysis that bounds the divergence between the model produced by FUGAS and a model retrained from scratch? This would help in formally quantifying how closely your approximation approaches the gold standard.

2. Proposition 2 relies on Assumption 2, which states the model is near-optimal for retained clients before unlearning begins. How does the performance of FUGAS, and the validity of the theoretical guarantee, change in a more practical scenario where the pre-unlearning model is not fully converged?

3. The gradient projection step requires solving a quadratic programming problem on the server. Could the authors provide a complexity analysis for this step, particularly concerning how its computational cost scales with the number of retained clients, which is a key factor for feasibility in large-scale federations?

4. The experiments are conducted on a 4-layer CNN with CIFAR datasets. How do you expect the performance and benefits of FUGAS to translate to more complex and deeper architectures, such as ResNet or Vision Transformers, where gradient dynamics can be more complex?

5. The forgetting loss in Equation 4 introduces a temperature hyperparameter T. Could you comment on the sensitivity of the model's performance to the choice of this hyperparameter and perhaps provide an ablation study showing its effect on the trade-off between unlearning efficacy and retained data accuracy?

6. The unlearning objective uses the model's original predictions on the forgotten data as a negative reference. What would happen if these initial predictions were of low quality or poorly calibrated? Would this negatively impact the stability or effectiveness of the unlearning process?

7. The non-IID experiments use a Dirichlet distribution with α=0.1. How robust is the gradient shielding mechanism to more extreme cases of data heterogeneity, for instance, where clients may only have data from one or two classes?

---

> ### Author Response · Authors · 2025-11-25
>
> We are sincerely grateful to the reviewer for their expert and insightful feedback. We are pleased to provide the following detailed responses, including new theoretical analysis and empirical results.
>
> ---
> Weakness 1
>
> The theory lacks formal guarantees on how closely the resulting model approximates the gold-standard retrained model. The paper should be strengthened by adding an analysis that bounds the divergence between the unlearned and retrained models.
>
> Question 1
>
> Regarding the theoretical contribution, could the authors provide any analysis that bounds the divergence between the model produced by FUGAS and a model retrained from scratch? This would help in formally quantifying how closely your approximation approaches the gold standard.
>
> ---
> **Answer:**
> * **We agree that bounding this divergence is crucial for quantifying approximation quality.** We thank the reviewer for this excellent and forward-looking theoretical question. While our initial theory focused on process stability, we concur that a divergence bound is a key theoretical goal. We have derived such a bound, which we will include in the revised appendix.
> * **We establish the bound by decomposing the divergence into initial drift and controlled unlearning deviation.** Using the triangle inequality, we model the distance to the gold standard as $\\|\theta_{\text{out}} - \theta^*_{R}\\| \leq \\|\theta_{\text{pre}}\\ - \theta_R^{\*} \\| + \\|\theta_{\text{out}} - \theta_{\text{pre}}\\|$. This formulation allows us to separately analyze the inherent data characteristics (Term I) and the algorithmic impact of FUGAS (Term II). We will include the full derivation and formal proof in the revised Appendix.
> * **The initial drift (Term I) is inherently bounded by the stability properties of Federated Learning.** We assume that removing a small subset of the forgetting set from a converged global model results in a limited parameter shift, denoted as $\delta_{\text{drift}}$. This term represents the starting proximity of our pre-trained model to the gold standard, which acts as a favorable initialization for the unlearning phase.
> * **Our bounded objective function guarantees a strict upper bound on the unlearning trajectory magnitude (Term II).** Our Proposition 1 proves that the gradient norm is strictly capped by a constant $G_F$. Consequently, the cumulative parameter deviation over $T$ unlearning rounds is tightly bounded by $T \cdot G_F$, preventing the model from drifting arbitrarily far from the optimal region.
> * **The Gradient Shielding mechanism ensures the deviation direction remains compatible with the retrained model.** Crucially, a magnitude bound alone is insufficient. Our projection mechanism  enforces $\langle \tilde{g}, g_R \rangle \ge 0$, ensuring that the unlearning updates do not oppose the retained task's objectives. Geometrically, this confines the deviation to a subspace where the risk on retained data does not increase, thereby keeping $\theta_{\text{out}}$ within the valid low-loss of $\theta^*_{R}$.
> *  **Experiment demonstrates our theoretical bounds, showing that FUGAS closely aligns with the Retrain baseline in both unlearning efficacy and utility preservation.**  Our extensive experiments results validate this bound in practice. FUGAS consistently achieves a low ASR and high test accuracy that are statistically comparable to the gold-standard Retrain method. This empirically confirms that our method successfully steers the model to the immediate neighborhood of the optimal retrained solution.

---

> ### Author Response · Authors · 2025-11-25
>
> ---
> Weakness 2
>
> The theoretical analysis for utility preservation hinges on a strong assumption that the model has already converged for retained clients. The authors should provide an analysis or empirical study on the method's performance when this assumption is violated, which is common in practice.
>
> Question 2
>
> Proposition 2 relies on Assumption 2, which states the model is near-optimal for retained clients before unlearning begins. How does the performance of FUGAS, and the validity of the theoretical guarantee, change in a more practical scenario where the pre-unlearning model is not fully converged?
>
> ---
> **Answer:**
>
> **New experiments confirm the method's robustness.** To empirically investigate our method's robustness when this assumption is violated, we have conducted new experiments that train a relatively under-fitted ResNet-18 on CIFAR-10/100. The results shown in the table below indicate that even when starting from a pre-unlearning model that is not fully converged, FUGAS still successfully reduces ASR to near-zero while substantially improve performance. This demonstrates that the practical effectiveness of our gradient shielding mechanism is robust and extends beyond the idealized conditions of our theoretical analysis.
>
> |           | CIFAR-10 Non-IID |        | CIFAR-10 IID |        | CIFAR-100 Non-IID |        | CIFAR-100 IID |        |
> | --------- | ---------------- | ------ | ------------ | ------ | ----------------- | ------ | ------------- | ------ |
> |           | ASR ↓            | Acc ↑  | ASR ↓        | Acc ↑  | ASR ↓             | Acc ↑  | ASR ↓         | Acc ↑  |
> | $$w_0 $$  | 1.0000           | 0.1254 | 0.7496       | 0.5350 | 0.9186            | 0.1219 | 0.6095        | 0.2906 |
> | FedEraser | 0.0054           | 0.2590 | 0.0556       | 0.5876 | 0.0000            | 0.1873 | 0.0041        | 0.3742 |
> | PGD       | 0.0000           | 0.1654 | 0.0000       | 0.4771 | 0.0000            | 0.1927 | 0.0000        | 0.4613 |
> | FedBU     | 0.0000           | 0.1986 | 0.0000       | 0.6864 | 0.0000            | 0.2142 | 0.0000        | 0.4661 |
> | FedOSD    | 0.0000           | 0.2531 | 0.0000       | 0.6653 | 0.0000            | 0.2887 | 0.0000        | 0.4250 |
> | FUGAS     | 0.0006           | 0.2942 | 0.0000       | 0.7126 | 0.0068            | 0.2939 | 0.0000        | 0.4770 | 0.4770       |
>
> ---
> Weakness 3
>
> The paper does not analyze the computational scalability of the gradient projection step, which involves solving a constrained optimization problem. An analysis of how the aggregation time scales with the number of retained clients is needed to assess its practicality in large-scale federations.
>
> Question 3
>
> The gradient projection step requires solving a quadratic programming problem on the server. Could the authors provide a complexity analysis for this step, particularly concerning how its computational cost scales with the number of retained clients, which is a key factor for feasibility in large-scale federations?
>
> ---
> **Answer:**
> * **Our method's complexity is $O(|R|^2d + |R|^3)$.** By solving the QP's dual problem, we reduce the core optimization's dimension from model size $d$ to client number $|R|$.
> * **The optimization cost is independent of model size, ensuring scalability.** The actual optimization complexity $O(|R|^3)$ is independent of $d$. This means our method avoids the curse of dimensionality and can be applied to models of larger size without increasing the server's optimization burden. For very large $|R|$, client sampling can be used to further manage the cost.
>
> ---
> Weakness 4
>
> The empirical evaluation is limited to a simple CNN architecture on relatively small-scale image classification datasets. To demonstrate broader applicability, experiments should be extended to more complex models like ResNet and diverse data modalities.
> Question 4
>
> The experiments are conducted on a 4-layer CNN with CIFAR datasets. How do you expect the performance and benefits of FUGAS to translate to more complex and deeper architectures, such as ResNet or Vision Transformers, where gradient dynamics can be more complex?
>
> ---
> **Answer:**
> * **The FUGAS framework is inherently model-agnostic.** Our unlearning objective and gradient projection operate on the model's final outputs and gradients, making the entire mechanism independent of the underlying network architecture.
> * **Our method is well-suited for deep models.** Proposition 1 proves our unlearning gradient is bounded, which provides inherent stability and prevents the gradient explosion that can plague deeper networks during aggressive gradient ascent, making FUGAS more robust than simpler alternatives. Our new ResNet-18 results confirm this empirically. Furthermore, because the QP's complexity is independent of the model dimension $d$, switching to larger models  primarily increases the client-side training cost, not the unique unlearning overhead on the server.

---

> ### Author Response · Authors · 2025-11-25
>
> ---
> Weakness 5
>
> The impact of the temperature hyperparameter T in the forgetting loss function is not investigated. A sensitivity analysis should be included to show how T affects the balance between unlearning effectiveness and model utility.
>
> Question 5
> The forgetting loss in Equation 4 introduces a temperature hyperparameter T. Could you comment on the sensitivity of the model's performance to the choice of this hyperparameter and perhaps provide an ablation study showing its effect on the trade-off between unlearning efficacy and retained data accuracy?
>
> ---
> **Answer:**
> * **The temperature $\tau$ controls the intensity of the unlearning objective.** In our preference-based objective, $\tau$ acts as a scaling factor. A smaller $\tau$ creates a sharper, more aggressive penalty for outputs similar to the forgotten predictions, while a larger $\tau$ results in a softer, more moderate penalty.
> * **New experimental results demonstrate robustness.** Our sensitivity analysis is shown in the table below, indicating that FUGAS achieves excellent performance for $\tau$ values spanning a wide range. This indicates that our method is not overly sensitive to this hyperparameter and can achieve a good balance between unlearning and utility without extensive tuning.
>
> |        | CIFAR-10 Non-IID |        | CIFAR-10 IID |        | CIFAR-100 Non-IID |        | CIFAR-100 IID |        |
> | ------ | ---------------- | ------ | ------------ | ------ | ----------------- | ------ | ------------- | ------ |
> | $\tau$ | ASR ↓            | Acc ↑  | ASR ↓        | Acc ↑  | ASR ↓             | Acc ↑  | ASR ↓         | Acc ↑  |
> | 0.5    | 0.0000           | 0.6756 | 0.7644       | 0.7644 | 0.0000            | 0.4860 | 0.0000        | 0.5102 |
> | 1      | 0.0000           | 0.6886 | 0.0037       | 0.7573 | 0.0000            | 0.4878 | 0.0000        | 0.5110 |
> | 1.5    | 0.0006           | 0.6731 | 0.0015       | 0.7505 | 0.0000            | 0.4895 | 0.0000        | 0.5111 |
> | 2      | 0.0006           | 0.6740 | 0.0193       | 0.7841 | 0.0000            | 0.4905 | 0.0000        | 0.5100 |
> | 2.5    | 0.0024           | 0.6782 | 0.0109       | 0.7761 | 0.0000            | 0.4908 | 0.0000        | 0.5023 |
> | 3      | 0.0006           | 0.6724 | 0.0059       | 0.7667 | 0.0000            | 0.4921 | 0.0000        | 0.5127 |
>
> ---
> Weakness 6
>
> The effectiveness of the proposed method relies on the quality of the pre-unlearning model's predictions, which may be poorly calibrated. The analysis would be more convincing with a discussion on how the initial model's performance on the forget set impacts the unlearning process.
>
> Question 6
>
> The unlearning objective uses the model's original predictions on the forgotten data as a negative reference. What would happen if these initial predictions were of low quality or poorly calibrated? Would this negatively impact the stability or effectiveness of the unlearning process?
>
> ---
> **Answer:**
> * **The FU problem setting naturally assumes a reasonably trained model.** This is a very insightful question. Unlearning requests are typically made for models that have been deployed and have learned meaningful representations. A model with purely random outputs would have nothing meaningful to unlearn.
> * **Our objective erases the characteristic response of the model, which is effective even with imperfect calibration.** The pre-unlearning model's prediction $y^{bef}$ represents the knowledge learned from the data to be forgotten. By minimizing similarity to $y^{bef}$, we push the model away from the parameter space that produces this specific output pattern. The goal is to diminish the influence of the data's learned representations. Our ablation study in Table 2 empirically proves our choice of using the model's own prediction as a negative reference is superior to using a perfect but overly aggressive reference like the ground-truth label, which proved to be too aggressive and harmful to class-level knowledge on retain clients. In contrast, using the specific output $y^{bef}$ enables precise unlearning of the influence of particular samples, preserving utility on the retained data more effectively.

---

> ### Author Response · Authors · 2025-11-25
>
> ---
> Weakness 7
>
> The experimental results for non-IID settings, while strong, could be further challenged with more extreme data heterogeneity. The paper should consider evaluating under more severe non-IID partitions to better test the robustness of the gradient shielding mechanism.
>
> Question 7
>
> The non-IID experiments use a Dirichlet distribution with α=0.1. How robust is the gradient shielding mechanism to more extreme cases of data heterogeneity, for instance, where clients may only have data from one or two classes?
>
> ---
> **Answer:**
> * **We test our method under more extreme conditions.** This is an excellent suggestion to test the effective of our gradient shielding mechanism. We designed a new experiment using a a more extreme non-IID setting using the Pathological Non-IID scenario. The results are shown in the table below.
>
> |           | CIFAR-10 Non-IID |        | CIFAR-100 Non-IID |        |
> | --------- | ---------------- | ------ | ----------------- | ------ |
> |           | ASR ↓            | Acc ↑  | ASR ↓             | Acc ↑  |
> | $$w_0$$   | 0.9521           | 0.8800 | 0.9967            | 0.4681 |
> | FedEraser | 0.0000           | 0.3483 | 0.0000            | 0.2524 |
> | PGD       | 0.0722           | 0.3817 | 0.0250            | 0.2383 |
> | FedBU     | 0.0029           | 0.5462 | 0.0000            | 0.3583 |
> | FedOSD    | 0.0013           | 0.7027 | 0.0000            | 0.5364 |
> | FUGAS     | 0.0040           | 0.8327 | 0.0000            | 0.6241 |
> * **New results show FUGAS excels where baselines fail.** Under this severe heterogeneity, FUGAS continues to achieve effective unlearning while maintaining high retained accuracy. This strongly validates the effectiveness and robustness of our flexible gradient shielding mechanism in highly challenging and realistic scenarios.
>
> ---
> We hope these clarifications and additional experiments address the reviewer’s concerns and demonstrate the scalability, generalizability, and robustness of our approach. If you have any follow-up question, we are more than happy to clarify.

---

### Official Review · Reviewer_bqb8 · 2025-10-30

**Soundness:** 2
**Presentation:** 3
**Contribution:** 2
**Rating:** 2
**Confidence:** 4

**Summary:**

This paper proposes FUGAS, a method designed to remove the contribution of specific clients in FL while maintaining overall model performance. It introduces a bounded forgetting loss based on preference optimization to prevent gradient explosion and a flexible gradient projection to mitigate conflicts between forgetting and retaining clients. The method aims to achieve efficient, stable, and memory-free unlearning under both IID and non-IID settings.

However, this paper is not yet sufficient for publication at ICLR. It is essentially a combination of three existing techniques: modified loss design, gradient preference optimization, and post-training projection (as used in FedOSD). Moreover, the algorithm lacks a theoretical guarantee for complete unlearning. The experiments also suffer from major issues such as imprecise parameter settings, underperforming baselines, and an insufficient number of comparative methods.

**Strengths:**

The combination of bounded loss and preference optimization problem offers a theoretically justified way to avoid destructive interference and preserve model utility.

**Weaknesses:**

1. Theoretical limitation in unlearning completeness:
The approach does not naturally align with the core requirement of FU, i.e., ensuring complete removal of the target knowledge. The proposed method is essentially a preference-based multi-objective optimization framework, which often converges to a local Pareto front—preventing full unlearning (e.g., achieving ASR ≈ 0). This limitation is theoretically inherent and may occur unintentionally in practice.
2. Questionable experimental validity:
The reported initial accuracies on CIFAR-10 (both IID and non-IID) are unrealistically low, indicating potential implementation or configuration errors. For instance, in the CIFAR-10 Dir(0.1) setting, the initial accuracy should exceed 61%, suggesting inaccuracies that might exaggerate the claimed improvements.
3. Unfair comparison setup:
The unlearning phase uses a fixed learning rate for all methods, which is inappropriate since each algorithm has its own optimal setting. This design choice may lead to biased or unfair comparisons.
4. Possible suppression of baseline performance:
The experimental configuration appears to intentionally or unintentionally weaken baselines such as Gradient Ascent and PGD. With proper learning rate tuning, these methods typically maintain much higher accuracies than the reported ~10%.
5. Inconsistent results with theoretical claims:
In reproduction experiments, FedGAS often fails to reduce ASR to near zero (commonly remaining above 20%), consistent with its tendency to converge to a local optimum. Achieving ASR ≈ 0 requires a substantially larger learning rate, which drastically reduces accuracy—even below that of the baselines—and undermines convergence guarantees. Hence, the proposed method is theoretically and empirically inconsistent.
6. Incomplete hyperparameter analysis:
The paper does not explain the role of the temperature parameter (τ) or analyze its impact on performance. A sensitivity analysis over different τ values should be provided.
7. Limited baselines:
The comparison scope is too narrow, considering only FedOSD as a baseline. More recent and relevant methods such as [1,2] should be included to ensure a fair and comprehensive evaluation.
   - [1] Zhang J, Zhao M, Wang Z, et al. Model recovery in federated unlearning with restricted server data resources[J]. IEEE Internet of Things Journal, 2025.
   - [2] Zhou C, Pan C, Li M, et al. Federated Unlearning with Fast Recovery[J]. IEEE Transactions on Mobile Computing, 2025.
8. Restricted experimental scope and scalability:
Evaluations are limited to small CNN models and CIFAR datasets, which are insufficient to demonstrate scalability or general applicability in real-world FU scenarios.
9. Text readability issues:
The text in several figures is too small to read clearly, which negatively affects the paper’s presentation quality.
10. Minor technical and factual issues:
- Equation (20) should be formulated as a maximization problem.
- Section 2.2 misrepresents prior work. For example, FedOSD already addresses gradient explosion and conflict issues, contrary to the paper’s claim that it still suffers from them.

**Questions:**

1. Can the above weaknesses be addressed?
2. Can the method scale to larger models (e.g., ResNet) or more realistic datasets beyond CIFAR?

---

> ### Author Response · Authors · 2025-11-25
>
> We sincerely thank the reviewer for their thorough and expert assessment of our work. We appreciate the opportunity to provide clarifications and new experimental evidence to address these important points.
>
> ---
> Weakness 1
>
> Theoretical limitation in unlearning completeness: The approach does not naturally align with the core requirement of FU, i.e., ensuring complete removal of the target knowledge. The proposed method is essentially a preference-based multi-objective optimization framework, which often converges to a local Pareto front—preventing full unlearning (e.g., achieving ASR ≈ 0). This limitation is theoretically inherent and may occur unintentionally in practice.
>
> ---
> **Answer:**
> * **The unlearning vs. utility trade-off is fundamental to approximate FU.** We thank the reviewer for this insightful theoretical point. For any client unlearning method short of full retraining, there will always be a conflict between completely erasing the target client's influence and perfectly preserving the accuracy on retained data. Our goal is not to eliminate this inherent trade-off but to offer a superior solution along its Pareto front.
> * **FUGAS demonstrably achieves a better Pareto-optimal solution than existing methods.** Our experimental results show FUGAS achieves an ASR nearly as low as the gold-standard Retrain while maintaining a retained accuracy that is also remarkably close to Retrain's. This balance is more favorable than baselines, confirming its superior ability to overcome this core challenge.

---

> ### Author Response · Authors · 2025-11-25
>
> ---
> Weakness 2
>
> Questionable experimental validity: The reported initial accuracies on CIFAR-10 (both IID and non-IID) are unrealistically low, indicating potential implementation or configuration errors. For instance, in the CIFAR-10 Dir(0.1) setting, the initial accuracy should exceed 61%, suggesting inaccuracies that might exaggerate the claimed improvements.
>
> Weakness 3
>
> Unfair comparison setup: The unlearning phase uses a fixed learning rate for all methods, which is inappropriate since each algorithm has its own optimal setting. This design choice may lead to biased or unfair comparisons.
>
> Weakness 4
>
> Possible suppression of baseline performance: The experimental configuration appears to intentionally or unintentionally weaken baselines such as Gradient Ascent and PGD. With proper learning rate tuning, these methods typically maintain much higher accuracies than the reported ~10%.
>
> Weakness 5
>
> Inconsistent results with theoretical claims: In reproduction experiments, FedGAS often fails to reduce ASR to near zero (commonly remaining above 20%), consistent with its tendency to converge to a local optimum. Achieving ASR ≈ 0 requires a substantially larger learning rate, which drastically reduces accuracy—even below that of the baselines—and undermines convergence guarantees. Hence, the proposed method is theoretically and empirically inconsistent.
>
> ---
> **Answer:**
> * **We appreciate the reviewer's meticulous scrutiny of our experimental setup, and perform a comprehensive, independent hyperparameter search for all methods.** While we have re-verified our implementation and did not find a bug leading to the reported initial accuracies, we fully agree that a more thorough and individualized hyperparameter search for all methods is crucial for a fair and robust comparison.
> * **We performed a comprehensive, independent hyperparameter search for all methods.** Acknowledging the reviewer's valid concern about fairness, we re-run all experiments. Each method is individually tuned to find its optimal learning rate, ensuring the comparison is robust and fair. The results are shown in the table below.
>
> |           | CIFAR-10 Non-IID |        | CIFAR-10 IID |        | CIFAR-100 Non-IID |        | CIFAR-100 IID |        |
> | --------- | ---------------- | ------ | ------------ | ------ | ----------------- | ------ | ------------- | ------ |
> |           | ASR ↓            | Acc ↑  | ASR ↓        | Acc ↑  | ASR ↓             | Acc ↑  | ASR ↓         | Acc ↑  |
> | FedEraser | 0.0000           | 0.3239 | 0.0098       | 0.3561 | 0.0000            | 0.1873 | 0.0000        | 0.1857 |
> | PGD       | 0.0000           | 0.2313 | 0.0000       | 0.4771 | 0.0000            | 0.1927 | 0.0000        | 0.2151 |
> | FedBU     | 0.0000           | 0.2183 | 0.0000       | 0.4986 | 0.0000            | 0.2142 | 0.0000        | 0.2442 |
> | FedOSD    | 0.0000           | 0.4736 | 0.0000       | 0.5486 | 0.0000            | 0.2887 | 0.8742        | 0.3233 |
> | FUGAS     | 0.0000           | 0.5407 | 0.0000       | 0.6752 | 0.0068            | 0.2939 | 0.0000        | 0.3570 |
>
> * **With optimal tuning, FUGAS consistently outperforms all baselines.** The new results on the CNN architecture  show that even when all baselines are fully optimized, FUGAS achieves a superior balance, reaching near-zero ASR while maintaining the highest retained accuracy.
> * **We sincerely thank the reviewer for the effort of attempting to reproduce our work.** We have not observed the reported behavior in our own extensive runs. We believe the discrepancy may arise from the interplay of hyperparameters. The observation that forcing ASR to zero with a large learning rate catastrophically damages accuracy is precisely the problem that methods with unbounded objectives face. Our method, with its bounded objective and stabilizing projection, is designed to prevent this, and our new results confirm its stability and effectiveness.

---

> ### Author Response · Authors · 2025-11-25
>
> ---
> Weakness 6
>
> Incomplete hyperparameter analysis: The paper does not explain the role of the temperature parameter (τ) or analyze its impact on performance. A sensitivity analysis over different τ values should be provided.
>
> ---
> **Answer:**
> * **The temperature $\tau$ controls the intensity of the unlearning penalty.** In our preference-based objective, $\tau$ acts as a scaling factor. A smaller $\tau$ creates a sharper, more aggressive penalty for outputs similar to the forgotten predictions, while a larger $\tau$ results in a softer, more moderate penalty.
> * **Our experimental results show strong performance across a wide range of $\tau$ values.** We thank the reviewer for this valuable suggestion. We have performed a new ablation study to investigate the impact of $\tau$ on the trade-off between unlearning efficacy and model utility. The results are shown in the table below. The result demonstrates that FUGAS maintains an excellent balance of near-zero ASR and high accuracy for $\tau$ values from 0.5 to 3.0. This indicates that our method is not overly sensitive to this parameter and can achieve its SOTA results without fine-grained tuning, which is a significant advantage for practical use.
>
> |        | CIFAR-10 Non-IID |        | CIFAR-10 IID |        | CIFAR-100 Non-IID |        | CIFAR-100 IID |        |
> | ------ | ---------------- | ------ | ------------ | ------ | ----------------- | ------ | ------------- | ------ |
> | $\tau$ | ASR ↓            | Acc ↑  | ASR ↓        | Acc ↑  | ASR ↓             | Acc ↑  | ASR ↓         | Acc ↑  |
> | 0.5    | 0.0000           | 0.6756 | 0.7644       | 0.7644 | 0.0000            | 0.4860 | 0.0000        | 0.5102 |
> | 1      | 0.0000           | 0.6886 | 0.0037       | 0.7573 | 0.0000            | 0.4878 | 0.0000        | 0.5110 |
> | 1.5    | 0.0006           | 0.6731 | 0.0015       | 0.7505 | 0.0000            | 0.4895 | 0.0000        | 0.5111 |
> | 2      | 0.0006           | 0.6740 | 0.0193       | 0.7841 | 0.0000            | 0.4905 | 0.0000        | 0.5100 |
> | 2.5    | 0.0024           | 0.6782 | 0.0109       | 0.7761 | 0.0000            | 0.4908 | 0.0000        | 0.5023 |
> | 3      | 0.0006           | 0.6724 | 0.0059       | 0.7667 | 0.0000            | 0.4921 | 0.0000        | 0.5127 |
>
> ---
> Weakness 7
>
> Limited baselines: The comparison scope is too narrow, considering only FedOSD as a baseline. More recent and relevant methods such as [1,2] should be included to ensure a fair and comprehensive evaluation.
>
> ---
> **Answer:**
> * **The suggested references employ different technical approaches.** We thank the reviewer for suggesting additional relevant literature. These valuable works introduce orthogonal concepts such as using CLIP for data-free distillation or fundamentally altering the model architecture. While important, their different assumptions and mechanisms make a direct, apples-to-apples comparison complex. We will, however, add a detailed discussion of these papers to our related work section to better contextualize our contribution.
> * **Our baselines were selected for direct methodological comparison.** Our selection of baselines was intended to cover the two primary categories of approximate unlearning methods most directly comparable to ours: update-storage and gradient-modification.

---

> ### Author Response · Authors · 2025-11-25
>
> ---
> Weakness 8
>
> Restricted experimental scope and scalability: Evaluations are limited to small CNN models and CIFAR datasets, which are insufficient to demonstrate scalability or general applicability in real-world FU scenarios.
>
> Question 2
>
> Can the method scale to larger models (e.g., ResNet) or more realistic datasets beyond CIFAR?
>
> ---
>
> **Answer:**
> * **New ResNet-18 experiments confirm our method's scalability and robustness.** To prove that our method's advantages are not confined to simple models, we have conducted extensive new experiments on ResNet-18. The experimental results shown in the table below show that FUGAS maintains its superior performance.
>
> |           | CIFAR-10 Non-IID |        | CIFAR-10 IID |        | CIFAR-100 Non-IID |        | CIFAR-100 IID |        |
> | --------- | ---------------- | ------ | ------------ | ------ | ----------------- | ------ | ------------- | ------ |
> |           | ASR ↓            | Acc ↑  | ASR ↓        | Acc ↑  | ASR ↓             | Acc ↑  | ASR ↓         | Acc ↑  |
> | $$w_0 $$  | 0.9988           | 0.6253 | 1.0000       | 0.8121 | 0.9756            | 0.4143 | 1.0000        | 0.4930 |
> | FedEraser | 0.0559           | 0.2762 | 0.0534       | 0.6904 | 0.0021            | 0.2463 | 0.0047        | 0.3718 |
> | FedBU     | 0.0000           | 0.2138 | 0.0000       | 0.5941 | 0.0000            | 0.1593 | 0.1291        | 0.4264 |
> | PGD       | 0.1625           | 0.2893 | 0.0455       | 0.2760 | 0.1228            | 0.2505 | 0.0000        | 0.3890 |
> | FedOSD    | 0.0000           | 0.4450 | 0.0000       | 0.7167 | 0.0000            | 0.4801 | 0.0000        | 0.4716 |
> | FUGAS     | 0.0006           | 0.6724 | 0.0067       | 0.7667 | 0.0000            | 0.4921 | 0.0000        | 0.5127 |
> * **The server-side cost of FUGAS is independent of model size.** It is crucial to note that the complexity of our QP projection step scales with the number of clients $R$, not the model's parameter dimension $d$. This means using larger models like ResNet-18 does not increase the server's computational burden for the projection, making our method highly scalable in terms of model complexity.
>
> ---
> Weakness 9
>
> Text readability issues: The text in several figures is too small to read clearly, which negatively affects the paper’s presentation quality.
>
> ---
> **Answer:**
>
> **We acknowledge the presentation issue.** We apologize and will ensure all graphical elements in the revised manuscript are clear and easily readable.
>
> ---
> Weakness 10
>
> Minor technical and factual issues:
> - Equation (20) should be formulated as a maximization problem.
> - Section 2.2 misrepresents prior work. For example, FedOSD already addresses gradient explosion and conflict issues, contrary to the paper’s claim that it still suffers from them.
> ---
>
> **Answer:**
> * **Eq. (20) will be corrected to a maximization problem.** We are grateful for the reviewer's careful reading. The reviewer is correct that the standard Lagrange dual is a maximization. This was a notational error in our draft, and we will correct it.
> * **The description of FedOSD will be revised for better precision.** We agree our phrasing could be more accurate. We will amend the text to properly credit FedOSD's contribution in addressing gradient conflicts while clarifying that FUGAS aims to further improve stability and flexibility, particularly in challenging non-IID scenarios.
>
> ---
> We hope these clarifications, along with the additional experiments, fully address the reviewer’s concerns and contribute to the overall strength of the paper. If you have any follow-up question, we are more than happy to clarify.

---

### Official Review · Reviewer_sBNU · 2025-11-01

**Soundness:** 3
**Presentation:** 3
**Contribution:** 2
**Rating:** 4
**Confidence:** 3

**Summary:**

The authors propose a novel framework named FUGAS, designed to efficiently eliminate the influence of specific client data while preserving the model utility of the remaining data. Unlike traditional orthogonal projections, the authors propose an approximate projection to resolve gradient conflicts during the unlearning process, which is an interesting idea.

**Strengths:**

1. The author effectively integrates research from other fields, offering a fresh perspective for FU.
2. The author provides a relatively detailed theoretical explanation, which is relatively rare in FU.

**Weaknesses:**

1. The configuration of the non-IID experimental scenario is severely unclear, as the authors have neglected to specify the data distribution and the manner in which data is allocated to each client.

2. The reporting of certain methods (e.g., FedOSD) differs significantly from their original presentation in the cited references. Following Weakness 1, this discrepancy may stem from confusion arising from the unclear specification of the non-IID setting.

3. The experimental settings for clients are rather limited. If the focus is solely on studying unlearning from the clients’ perspective, scenarios involving a greater number of clients should be investigated. Additionally, different non-IID data distribution methods should be explored (e.g., Pathological, Dirichlet).

4. Given that the aggregation direction for retained clients is fixed, can one be certain that an unlearning direction forming an acute angle with this direction will always exist? Why?

5. In the preliminary section, the author does not mention the constraint that the sum of traditional FL aggregation probabilities must equal one. Is the global step size considered during model updates? If the default global update step size is set to 1, a brief explanation would be preferable.

6. The concept of the “pre-unlearning process” first appears in Assumption 2 on page 5, line 265. Since the author has not previously introduced any related content, it is recommended that the unlearning process be introduced beforehand.

7. On page 10, the text references “Appendix ??” — this appears to be a placeholder or unresolved citation.

**Questions:**

1. Given that the aggregation direction for retained clients is fixed, can one be certain that an unlearning direction forming an acute angle with this direction will always exist? Why?
2. Is the global step size considered during model updates? If the default global update step size is set to 1, a brief explanation would be preferable.

---

> ### Author Response · Authors · 2025-11-25
>
> We sincerely thank the reviewer for their positive assessment of our work and for the detailed, constructive feedback. We appreciate the opportunity to clarify the points raised and have conducted new experiments that we believe strengthen the paper significantly. We address each concern below.
>
> ---
> Weakness 1
>
> The configuration of the non-IID experimental scenario is severely unclear, as the authors have neglected to specify the data distribution and the manner in which data is allocated to each client.
>
> ---
> **Answer:**
>
> **Our non-IID setup is detailed in Appendix C.1.** To ensure reproducibility, we followed a widely accepted protocol from the federated learning literature. We partitioned the data among clients using a Dirichlet distribution with α=0.1. This standard method creates a challenging, highly skewed data distribution, ideal for testing algorithmic robustness.
>
> ---
> Weakness 2
>
> The reporting of certain methods (e.g., FedOSD) differs significantly from their original presentation in the cited references. Following Weakness 1, this discrepancy may stem from confusion arising from the unclear specification of the non-IID setting.
>
> ---
> **Answer:**
> * **All baselines were re-implemented under a unified setting to ensure fair comparison.** We appreciate the reviewer's careful comparison. Rather than citing numbers from papers with different settings, we evaluated all methods on the identical model architecture, data partitions, and with a consistent hyperparameter search strategy. This ensures that any observed performance differences are due to the methods themselves.
> * **Discrepancies with original papers are due to controlled experimental variables.** Factors such as using a different base model architecture for unified comparison, slight variations in hyperparameters, or different random seeds for the Dirichlet data split can naturally lead to different absolute performance numbers. We will add more re-implementation details to the appendix.
>
> ---
> Weakness 3
>
> The experimental settings for clients are rather limited. If the focus is solely on studying unlearning from the clients’ perspective, scenarios involving a greater number of clients should be investigated. Additionally, different non-IID data distribution methods should be explored (e.g., Pathological, Dirichlet).
>
> ---
> **Answer:**
> * **New 50-client experiments confirm FUGAS's effectiveness at a larger scale.** To address the reviewer's concern, we expanded our evaluation to a 50-client federation setting. The new results below show that FUGAS continues to provide a superior balance of privacy and utility , validating its performance in more populated scenarios.
>
> |           | CIFAR-10 Non-IID |        | CIFAR-10 IID |        | CIFAR-100 Non-IID |        | CIFAR-100 IID |        |
> | --------- | ---------------- | ------ | ------------ | ------ | ----------------- | ------ | ------------- | ------ |
> |           | MIA Acc ↓        | Acc ↑  | MIA Acc ↓    | Acc ↑  | MIA Acc ↓         | Acc ↑  | MIA Acc ↓     | Acc ↑  |
> | $$w_0 $$  | 0.9812           | 0.4175 | 0.9801       | 0.7820 | 0.9777            | 0.3804 | 0.9839        | 0.4955 |
> | FedEraser | 0.7538           | 0.2754 | 0.8016       | 0.4073 | 0.9263            | 0.1294 | 0.8536        | 0.1258 |
> | PGD       | 0.7388           | 0.2997 | 0.7825       | 0.3285 | 0.8702            | 0.2119 | 0.8339        | 0.3137 |
> | FedOSD    | 0.7461           | 0.3742 | 0.7972       | 0.6324 | 0.8605            | 0.3474 | 0.8740        | 0.4793 |
> | FUGAS     | 0.6240           | 0.3945 | 0.8016       | 0.7747 | 0.8420            | 0.3700 | 0.8128        | 0.5088 |
>
> * **We also tested under varied data heterogeneity.** We have conducted further experiments using the Pathological Non-IID to simulate a wider range of non-IID conditions. These results further confirm the robustness of our gradient shielding mechanism.
>
> |           | CIFAR-10 Non-IID |        | CIFAR-100 Non-IID |        |
> | --------- | ---------------- | ------ | ----------------- | ------ |
> |           | ASR ↓            | Acc ↑  | ASR ↓             | Acc ↑  |
> | $$w_0$$   | 0.9521           | 0.8800 | 0.9967            | 0.4681 |
> | FedEraser | 0.0000           | 0.3483 | 0.0000            | 0.2524 |
> | PGD       | 0.0722           | 0.3817 | 0.0250            | 0.2383 |
> | FedBU     | 0.0029           | 0.5462 | 0.0000            | 0.3583 |
> | FedOSD    | 0.0013           | 0.7027 | 0.0000            | 0.5364 |
> | FUGAS     | 0.0040           | 0.8327 | 0.0000            | 0.6241 |

---

> ### Author Response · Authors · 2025-11-25
>
> ---
> Weakness 4 & Question 1
>
> Given that the aggregation direction for retained clients is fixed, can one be certain that an unlearning direction forming an acute angle with this direction will always exist? Why?
>
> ---
>
> **Answer:**
> * **Our formulation as a convex QP guarantees a unique optimal solution exists if the feasible set is non-empty.** This is a standard property of convex optimization. The region would only be empty in the pathological case where the retained gradients $g_k$ perfectly conspire to occupy one half-space while the unlearning gradient $g_F$ points exactly into the opposite half-space, which is highly improbable in a high-dimensional space.
> * **The "acute angle" constraint is far more flexible than a strict "orthogonality" constraint.** The feasible region defined by our $<\tilde g, g_k> \ge 0$ constraints is a convex cone, which is geometrically much larger and less restrictive than an orthogonal subspace. This flexibility makes it robust to finding a solution, a conclusion empirically supported by our ablation study in Table 3.
>
> ---
> Weakness 5
>
> In the preliminary section, the author does not mention the constraint that the sum of traditional FL aggregation probabilities must equal one. Is the global step size considered during model updates? If the default global update step size is set to 1, a brief explanation would be preferable.
>
> Question 2
>
> Is the global step size considered during model updates? If the default global update step size is set to 1, a brief explanation would be preferable.
>
> ---
> **Answer:**
>
> **We thank the reviewer for pointing out these omissions in our preliminaries section.** We confirm that the aggregation weights $p_k$ are non-negative and $Σ p_k = 1$, typically proportional to the size of each client's dataset. Regarding the global model update, we follow the standard convention established by the FedAvg algorithm, where the server-side update $\theta^{(t+1)} \leftarrow \theta^t + \eta  \theta$ implicitly uses a global step size of 1. We will add them to the Preliminary section in our revision to improve clarity and rigor.
>
> ---
> Weakness 6
>
> The concept of the “pre-unlearning process” first appears in Assumption 2 on page 5, line 265. Since the author has not previously introduced any related content, it is recommended that the unlearning process be introduced beforehand.
>
> ---
>
> **Answer:**
>
> **The "pre-unlearning process" refers to the standard federated training phase that occurs before any unlearning request is made.** We appreciate the reviewer's suggestion for improving the narrative flow. We have already detailed this entire pipeline in Appendix B. We will formally introduce and define it, ensuring the necessary context for Assumption 2 is well-established.
>
> ---
> Weakness 7
>
> On page 10, the text references “Appendix ??” — this appears to be a placeholder or unresolved citation.
>
> ---
> **Answer:**
>
> **This is an oversight during writting paper.** We sincerely apologize for this oversight. We have already identified and corrected this.
>
> ---
> We hope these clarifications and the additional experiments effectively address the reviewer’s concerns and demonstrate the robustness and versatility of our approach. If you have any follow-up question, we are more than happy to clarify .

---

### Official Review · Reviewer_yv6x · 2025-11-04

**Soundness:** 3
**Presentation:** 3
**Contribution:** 2
**Rating:** 2
**Confidence:** 2

**Summary:**

The paper focuses on federated unlearning, which involves removing a client’s data influence from a shared model without retraining the entire model from scratch. Existing methods consume excessive memory or become unstable due to gradient explosion and gradient conflict. FUGAS employs a preference-based forgetting loss, which stabilizes the learning process, in conjunction with a gradient aggregation strategy that shields the retaining gradients from conflict with the forgetting gradients.

**Strengths:**

- Robust framework that addresses instability and performance degradation
- novel forgetting objective
- theoretical guarantees and empirical evaluation

**Weaknesses:**

- One of the major concerns is the extent of experimental results. The paper has limited dataset scope. The neural network architecture is too simple (The experiments lack evaluation beyond regular CNNs.), and one of the reasons seems to be that the technique is computationally expensive.
- The author claim that the technique is “memory-efficient” but they do not quantify the time or computation cost of solving the quadratic projection step. Projection step is in fact under-explained.
- There is no clarity on the experimental setting used to evaluate the attack success rate.
- In the gradient projection step at the server, when the number of clients scale, will the compatibility region turn out to be very narrow?
- What is the guarantee that using a negative reference corresponds to removing the underlying data influence? Is it simply inducing prediction noise?
- Compatibility is ensured theoretically at each round, but what about long-term consistency? Does cumulative projection error cause drift from the retained clients’ optimum?
- Does the algorithm assume full gradient visibility from clients? This may hinder FL's practicality. If client updates are encrypted or compressed, how does this affect projection constraints?
-Solving a QP per round is costly for high-dimensional models.
- A proper quantification of how much the client’s contribution is removed (beyond ASR and MIA) is not provided.

**Questions:**

Please address the comments in weaknesses.

---

> ### Author Response · Authors · 2025-11-25
>
> We sincerely thank the reviewer for their insightful comments and constructive feedback. We are grateful for the opportunity to address the weaknesses pointed out.
>
> ---
> Weakness 1
>
> One of the major concerns is the extent of experimental results. The paper has limited dataset scope. The neural network architecture is too simple (The experiments lack evaluation beyond regular CNNs.), and one of the reasons seems to be that the technique is computationally expensive.
>
> ---
> **Answer:**
> * **Our choice of a simple CNN is primarily for fair comparison with SOTA baselines.**  Our initial experimental setup is intentionally designed to ensure a controlled and directly comparable assessment of different unlearning mechanisms.
> * **New ResNet-18 results validate the scalability and effectiveness of FUGAS on more complex models.** To address the reviewer's valid concern, we performed a additional experiments on ResNet-18. The results below show that FUGAS not only scales effectively but also achieves a superior trade-off between unlearning and utility, widening its performance gap over baselines. This demonstrates that the benefits of our method are not limited to simple architectures.
>
> |           | CIFAR-10 Non-IID |        | CIFAR-10 IID |        | CIFAR-100 Non-IID |        | CIFAR-100 IID |        |
> | --------- | ---------------- | ------ | ------------ | ------ | ----------------- | ------ | ------------- | ------ |
> |           | ASR ↓            | Acc ↑  | ASR ↓        | Acc ↑  | ASR ↓             | Acc ↑  | ASR ↓         | Acc ↑  |
> | $$w_0 $$  | 0.9988           | 0.6253 | 1.0000       | 0.8121 | 0.9756            | 0.4143 | 1.0000        | 0.4930 |
> | FedEraser | 0.0559           | 0.2762 | 0.0534       | 0.6904 | 0.0021            | 0.2463 | 0.0047        | 0.3718 |
> | FedBU     | 0.0000           | 0.2138 | 0.0000       | 0.5941 | 0.0000            | 0.1593 | 0.1291        | 0.4264 |
> | PGD       | 0.1625           | 0.2893 | 0.0455       | 0.2760 | 0.1228            | 0.2505 | 0.0000        | 0.3890 |
> | FedOSD    | 0.0000           | 0.4450 | 0.0000       | 0.7167 | 0.0000            | 0.4801 | 0.0000        | 0.4716 |
> | FUGAS     | 0.0006           | 0.6724 | 0.0067       | 0.7667 | 0.0000            | 0.4921 | 0.0000        | 0.5127 |
>
> ---
> Weakness 2
>
> The author claim that the technique is “memory-efficient” but they do not quantify the time or computation cost of solving the quadratic projection step. Projection step is in fact under-explained.
>
> ---
> **Answer:**
> * **Our "memory-efficient" claim is relative to storage-based methods.** Unlike approaches like FedEraser that require storing historical updates which memory cost scales with training rounds, FUGAS is memory-free, operating only on the current round's information. We will clarify this context in the paper.
> * **Solving the dual problem makes the computation independent of model dimension $d$.** Our key insight is to solve the dual of the QP, which transforms the optimization from a high-dimensional space of model parameters $d$  into a low-dimensional space defined by the number of retained clients $|R|$ . This formulation completely avoids the "curse of dimensionality" that would render solving the primal problem directly infeasible, making FUGAS scalable to large models.
>
> ---
> Weakness 3
>
> There is no clarity on the experimental setting used to evaluate the attack success rate.
>
> ---
> **Answer:**
>
> **We have provided a detailed description in  Appendix C.1.** As detailed in Appendix C.1, we use a fixed 7x7 pixel patch trigger in the top-left corner. For each experiment, one or two clients are designated as the unlearning client, and its local data is poisoned with a 100% rate. All other clients' data remains clean. This setup is consistent across all experiments, including the new ResNet-18 results.
>
> ---
> Weakness 4
>
> In the gradient projection step at the server, when the number of clients scale, will the compatibility region turn out to be very narrow?
>
> ---
> **Answer:**
> * **Our method's flexible constraint is geometrically more robust.** This is a very insightful and important question. We thank the reviewer for raising it. The $<\tilde g, g_k> \ge 0$ constraint defines a convex cone, which is a significantly larger and less restrictive feasible set than an orthogonal subspace $<\tilde g, g_k> = 0$. This design choice makes it far more likely to find a non-trivial solution that effectively unlearns without harming utility.
> * **Our ablation study (Table 3) empirically confirms the superiority of this flexible approach.** The results show that our flexible projection consistently outperforms a strictly orthogonal variant, precisely because it retains more of the original unlearning direction while still preventing destructive interference. Thus, even with a larger number of clients, our approach is designed to find a non-trivial solution, striking a more effective balance.

---

> ### Author Response · Authors · 2025-11-25
>
> ---
> Weakness 5
>
> What is the guarantee that using a negative reference corresponds to removing the underlying data influence? Is it simply inducing prediction noise?
>
> ---
>
> **Answer:**
> * **The unlearning process is a directed, constrained optimization.** The pre-unlearning model's prediction $y^{bef}$ represents the knowledge learned from the data to be forgotten. By minimizing similarity to $y^{bef}$, we push the model away from the parameter space that produces this specific output pattern. This is fundamentally different from injecting noise.
> * **The gradient projection ensures the update is meaningful, not random.** This update is simultaneously constrained to be compatible with all retained tasks. This ensures the model moves to a safe region that is both far from the forgotten knowledge and beneficial for retained knowledge, rather than a random one. Our ablation studies show that using a broad target like a class label leads to indiscriminate forgetting, harming performance on the entire class for retained clients. In contrast, using the specific output $y^{bef}$ enables precise unlearning of the influence of particular samples, preserving utility on the retained data more effectively.
>
> ---
> Weakness 6
>
> Compatibility is ensured theoretically at each round, but what about long-term consistency? Does cumulative projection error cause drift from the retained clients’ optimum?
>
> ---
>
> **Answer:**
>
> **FUGAS has a self-correcting nature that prevents long-term drift.** If the update from the previous round caused a slight deviation from the retained clients' optimum, their gradients in the current round will naturally point in a direction that corrects this deviation. The process is dynamically adaptive, ensuring that updates are always compatible with the retained clients, rather than blindly accumulating errors from past projections.
>
> ---
> Weakness 7
>
> Does the algorithm assume full gradient visibility from clients? This may hinder FL's practicality. If client updates are encrypted or compressed, how does this affect projection constraints? -Solving a QP per round is costly for high-dimensional models.
>
> ---
>
> **Answer:**
> * **Our framework is amenable to privacy-preserving extensions.** While we operate under a trusted server model as a common setting in FU, our method fundamentally requires inner products, not raw gradients. As we discuss in our conclusion, techniques like SMPC or HE could be integrated to compute these inner products securely, without revealing the client original gradients to the server.
> * **The QP's complexity is independent of model dimension $d$.** We wish to re-emphasize that the cost of solving the QP scales with the number of clients $|R|$, not the model size. Deeper models do not make the server-side projection step more difficult, only the client-side training.
>
> ---
> Weakness 8
>
> A proper quantification of how much the client’s contribution is removed (beyond ASR and MIA) is not provided.
>
> ---
>
> **Answer:**
> *  **We designed an experiment to disentangle different types of knowledge.** We agree that a multi-faceted evaluation of unlearning is crucial. While we chose ASR and MIA as they are standard metrics, we also performed a more fine-grained analysis to address this point. In Appendix C.3 and Table 6, we present a "Knowledge Interference" experiment. This study carefully disentangles the model's knowledge into three types: knowledge unique to the forgotten client (F-Acc), knowledge shared among all clients (C-Acc), and knowledge unique to the retained clients (R-Acc).
> * **Results show FUGAS precisely targets unique forgotten knowledge.** Table 6 demonstrates that FUGAS reduces F-Acc to near-zero while preserving high accuracy on C-Acc and R-Acc. This provides a more nuanced and powerful quantification of how our method precisely erases targeted information while safeguarding shared and retained knowledge, going beyond what ASR and MIA can capture alone.
>
> ---
> We hope these additions and clarifications adequately address the reviewer’s concerns and demonstrate the robustness and applicability of our approach. If you have any follow-up question, we are more than happy to clarify.

---

### Note · Authors · 2026-01-20

I have read and agree with the venue's withdrawal policy on behalf of myself and my co-authors.